# MULTIPLE IMAGES DISTRACT LARGE MULTIMODAL MODELS VIA ATTENTION FRAGMENTATION

## ABSTRACT

Many everyday tasks involve integrating information across multiple images, such as comparing photos and reading social media posts. Recent Large Multimodal Models (LMMs) therefore accept multiple images, yet open-source models remain far from reliable in multi-image understanding, with accuracies often falling below 50% on recent evaluations. We analyze how these models allocate attention across images when visual tokens are processed in a single autoregressive, causally masked sequence. Our study uncovers a joint failure mode: the same background positions in each image repeatedly attract high attention while contributing little to prediction, and this effect is stronger for earlier images due to one-way attention under causal masking. We term this phenomenon attention fragmentation, as attention is split across non-informative tokens instead of binding evidence between images. These high-attention, low-utility tokens correspond to attention sinks previously observed in LLMs. To address attention fragmentation, we introduce Attention Remasking (AR), a post-training edit that operates on attention scores where the causal mask is enforced. AR masks sink tokens column-wise to prevent any query from attending to them, and selectively unmasks cross-image visual tokens deemed relevant by a grounded patch relevance score. The attention freed from the masked sinks is reassigned to these unmasked links, creating forward connections from earlier to later images while preserving text autoregression. AR reduces attention fragmentation and improves accuracy over post-training baselines on recent multi-image benchmarks, delivering more effective cross-image integration without additional training.

## 1 INTRODUCTION

Humans easily draw insight from multiple images, whether comparing photos of similar items, browsing social media posts, or following visual instructions. Motivated by these natural use cases, recent Large Multimodal Models (LMMs) have begun to accept multiple images via visual tokens, enabling reasoning across images rather than treating each in isolation (Jiang et al., 2024; Li et al., 2025). To probe these emerging capabilities, researchers have introduced multi-image evaluation benchmarks that test skills such as comparison, retrieval, scene and temporal understanding, and description writing (Zhao et al., 2024; Liu et al., 2024a). Despite progress in modeling multiple images, the results from benchmarks show underperformance of models, with leading proprietary models reaching only about 55–68% accuracy overall. Open-source models have lagged behind, for example, achieving below 50% in MMIU (Meng et al., 2025) and below 35% accuracy in MuirBench (Wang et al., 2025), indicating that integrating information across images remains a key challenge.

Current LMMs often handle multi-image inputs by incorporating them as visual tokens within the same autoregressive sequence used for text, so each query token attends over a long mixture of image and text keys (Jiang et al., 2024; Li et al., 2025). As shown in Fig. 1, our empirical analysis of this setting reveals a recurring attention pattern: within each image, a subset of visual tokens consistently absorbs a disproportionate amount of attention despite contributing little to the model's predictions. These tokens recur at similar background locations across images and are more pronounced in earlier images. These tokens are attention sinks, a structural adaptation to softmax normalization that stores excess attention scores without contributing to value computation (Gu et al., 2025). Attention sink has been found in the first tokens in text and the background tokens in images (Kang et al., 2025). The stronger sinks in earlier images are correlated with the causal masking on the attention

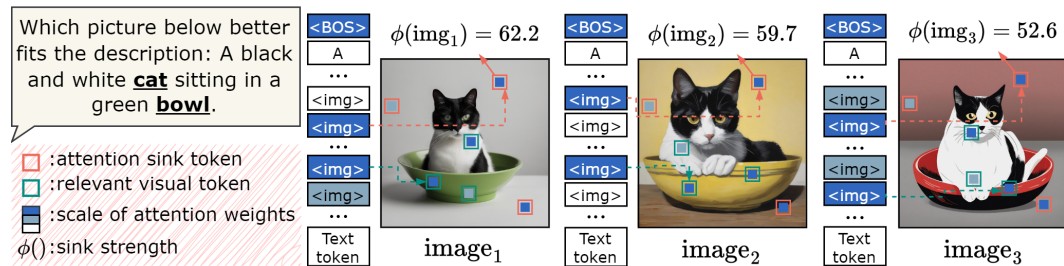

Figure 1: **Attention Fragmentation.** In multi-image LMMs, attention concentrates on attention sink tokens recurring in similar background positions in each image, with stronger sinks in earlier images. This diverts attention from task-relevant cues and fragments cross-image focus.

scores, where tokens are only allowed to attend to previous tokens and not future ones, so earlier images are exposed to more queries and accumulate more sink mass, resulting in a biased distribution of attention weights. We found that when multiple images are present, this combination of repeated sinks and uneven attention allocation fragments the model's focus; instead of focusing on task-relevant cross-image cues, the model spreads its attention thinly across these repetitive, non-informative anchors. We refer to this compound issue as attention fragmentation, a phenomenon that limits multi-image understanding in current LMMs. We observed that attention fragmentation is associated with subpar performance and the recency bias, in which later images disproportionately influence predictions and answers change with image order (Tian et al., 2025).

To address this phenomenon, we propose Attention Remasking (AR), a post-training edit that operates directly on the attention scores, where the causal masks are applied in the LLM decoder. AR improves multi-image understanding by masking sink keys and unmasking task-relevant cross-image keys, so that attention can flow across relevant visual tokens rather than being absorbed by repeated sinks. To prevent the effect from repeated sinks across images, AR masks the identified sink keys column-wise across all queries, which prevents sink tokens from being attended by other tokens. To enable cross-image relationships that are otherwise blocked by causal masking, AR modifies only the visual part of the mask to unmask attention from earlier queries to later image tokens, while preserving text autoregression. Because masked attention scores were never trained and are unreliable, AR assigns freed attention scores from masked sink keys to the newly unmasked keys proportional to grounded patch-level CLIP score (Radford et al., 2021; Hessel et al., 2021). We use Grounding DINO (Liu et al., 2024b) to propose regions related to the task instruction in each image and map those regions to the model's visual tokens. For tokens inside these regions, we compute the cosine similarity between each token's CLIP patch embedding and the instruction embedding, then normalize the results into a token-wise relevance distribution. By masking sinks and unmasking task-relevant visual attention, AR addresses attention fragmentation as one coherent problem, reducing distractions and restoring cross-image focus without retraining or hyperparameter tuning.

Our contributions are as follows: (1) We identify attention fragmentation in multi-image LMMs: repeated background sinks attract high attention across images, with stronger sinks on earlier images, which suppresses cross-image integration and induces order sensitivity, supported by analyses such as the symmetric Chamfer distance and image-level entropy. (2) We propose Attention Remasking (AR), a post-training edit at the pre-softmax stage that masks sink tokens and unmasks a sparse set of cross-image visual tokens deemed relevant by grounded patch-level CLIP score from the CLIP's visual encoder, while preserving text autoregression. (3) We demonstrate improved multi-image understanding on multi-image benchmarks. AR increases accuracy and reduces sensitivity to image order, outperforming post-training baselines without retraining or hyperparameter tuning.

## 2 PRELIMINARIES

**Attention mechanism.** We consider a decoder-based Large Multmodal Model (LMM) with $L$ layers and hidden width $D$. An input consists of text and $M$ images, tokenized into a single sequence of length $N$. For each image $m \in \{1, \ldots, M\}$, let $\mathcal{V}_m \subset \{1, \ldots, N\}$ be the indices of its visual tokens, and let $\mathcal{V} = \bigcup_{m=1}^{M} \mathcal{V}_m$ be the set of all visual-token indices. Let $\mathcal{T} \subset \{1, \ldots, N\}$ denote

the text-token indices. The $j$-th input token to the layer $\ell$ is $x_j^{\ell-1} \in \mathbb{R}^D$, and the stacked states form $X^{\ell-1} \in \mathbb{R}^{N \times D}$. Queries and keys are computed in the standard way, $Q^\ell = X^{\ell-1} W_Q^\ell$ and $K^\ell = X^{\ell-1} W_K^\ell$, with $W_Q^\ell, W_K^\ell \in \mathbb{R}^{D \times d_k}$ and key dimension $d_k$. The attention score matrix is

$$Z^\ell = \frac{Q^\ell K^{\ell\top}}{\sqrt{d_k}} + \mathcal{M}_{\text{causal}}, \tag{1}$$

where $\mathcal{M}_{\text{causal}} \in \mathbb{R}^{N \times N}$ is the causal mask with $\mathcal{M}_{\text{causal}, i,j} = -\infty$ for $i < j$ and $0$ otherwise. Row-wise softmax yields attention weights

$$\alpha_{i,j}^\ell = \text{softmax}\big(Z_{i,:}^\ell\big)_j, \qquad \sum_{j \leq i} \alpha_{i,j}^\ell = 1. \tag{2}$$

**Attention sinks.** Prior work reports that Transformer models can allocate large attention to tokens with little semantic value, a behavior termed attention sink, and attributes it to unusually large activations in a small set of hidden dimensions together with the softmax normalization (Gu et al., 2025). In LLMs, sinks often occur at fixed positions such as the first token; in LMMs, they appear on visual background tokens (Kang et al., 2025). To identify visual sink tokens, we follow the dimension-based criterion used in previous literature (Gu et al., 2025; Kang et al., 2025). Let $\mathcal{D}_{\text{sink}} \subset \{1, \ldots, D\}$ be the indices of potential sink dimensions. We estimate per-dimension statistics by passing the calibration corpus to the LLM, for each $d \in \mathcal{D}_{\text{sink}}$; let $\mu_d$ and $\sigma_d$ denote the mean and standard deviation of the $d$-th hidden coordinate measured on the calibration corpus (Sun et al., 2024a). For any hidden state $x \in \mathbb{R}^D$, define the sink score

$$\phi(x) = \max_{d \in \mathcal{D}_{\text{sink}}} \frac{x[d] - \mu_d}{\sigma_d}. \tag{3}$$

A token is flagged as a sink at layer $\ell$ if its previous-layer state satisfies $\phi\big(x_j^{\ell-1}\big) \geq \tau$, where $\tau$ is the threshold used in sink-identification work. For image $m$, the set of visual sink tokens at layer $\ell$ is

$$\mathcal{S}_m^\ell = \Big\{ j \in \mathcal{V}_m : \phi\big(x_j^{\ell-1}\big) \geq \tau \Big\}, \qquad \mathcal{S}^\ell = \bigcup_{m=1}^M \mathcal{S}_m^\ell. \tag{4}$$

This procedure isolates tokens that attract high attention due to massive activation in sink dimensions while contributing little to value computation, as documented for both text and visual sinks.

## 3 ATTENTION FRAGMENTATION

In Large Multimodal Models (LMMs), multi-image inputs are processed in a single, causally masked sequence, and attention weights are computed via a row-wise softmax over keys (Li et al., 2025; Jiang et al., 2024). Prior work shows that softmax normalization and optimization dynamics can induce attention sinks, tokens that absorb excess attention while contributing little to value computation (Gu et al., 2025). Building on this, we empirically examine how attention is allocated across images within the same autoregressive context in LMMs.

### 3.1 EMPIRICAL OBSERVATION OF ATTENTION FRAGMENTATION

**Repeated visual sinks at matched background positions.** In multi-image inputs, we observe that per-image sink tokens $\mathcal{S}_m^\ell$ recur in similar spatial regions across images within the same example, drawing substantial attention despite low semantic utility. To quantify the effect, we represent each visual token on the 2-D ViT patch grid of size $R \times C$: token $j$ at row–column $(r, c)$ is mapped to normalized image coordinates $\mathbf{u}_j = (r/R, c/C) \in [0, 1]^2$; distances are therefore purely spatial in image coordinates. For any unordered image pair $(m, m')$ in the same multi-image example with sink sets $\mathcal{S}_m^\ell = \{\mathbf{u}\}$ and $\mathcal{S}_{m'}^\ell = \{\mathbf{u}'\}$, we measure region-level repetition using the *symmetric Chamfer distance* on the 2-D token locations,

$$d_{\text{Chamfer}}(\mathcal{S}_m^\ell, \mathcal{S}_{m'}^\ell) = \frac{1}{|\mathcal{S}_m^\ell|} \sum_{\mathbf{u} \in \mathcal{S}_m^\ell} \min_{\mathbf{u}' \in \mathcal{S}_{m'}^\ell} \|\mathbf{u} - \mathbf{u}'\|_2 + \frac{1}{|\mathcal{S}_{m'}^\ell|} \sum_{\mathbf{u}' \in \mathcal{S}_{m'}^\ell} \min_{\mathbf{u} \in \mathcal{S}_m^\ell} \|\mathbf{u}' - \mathbf{u}\|_2, \tag{5}$$

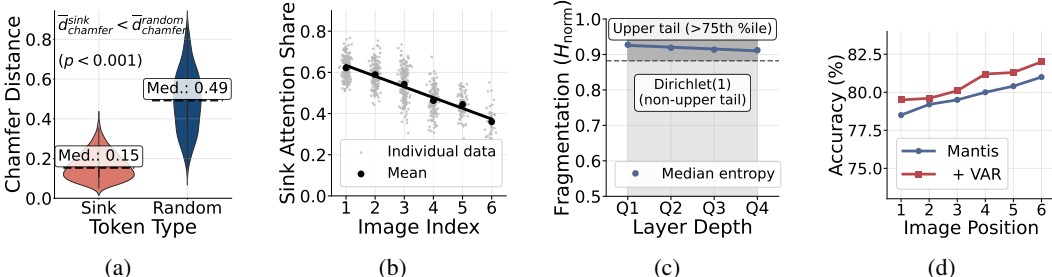

(a)             (b)             (c)             (d)

Figure 2: **Empirical analysis of attention fragmentation.** (a) *Repeated sinks across images:* symmetric Chamfer distance (Eqs. (5) and (6)) between per-image sink sets vs. random baseline. (b) *Positional skew:* sink attention share $s_m$ by image index $m$ (Eq. (7)); earlier images absorb more sink mass. (c) *Level of fragmentation:* normalized entropy $H_{\mathrm{norm}}$ across depth (Eq. (8)); values remain high relative to a Dirichlet(1) null, indicating persistent fragmentation. (d) *Positional bias:* position-wise accuracy under shows recency bias and persists after visual attention distribution (VAR).

where lower values indicate stronger spatial recurrence. For an example with $M$ images, we summarize recurrence by the *per-example average* over all unordered pairs

$$\overline{d}^{\ell}_{\mathrm{Chamfer}} \;=\; \frac{2}{M(M-1)} \sum_{1 \le m < m' \le M} d_{\mathrm{Chamfer}}(\mathcal{S}^{\ell}_m, \mathcal{S}^{\ell}_{m'}). \tag{6}$$

As shown in Fig. 2a, the per-example average $\overline{d}^{\ell}_{\mathrm{Chamfer}}$ across images in the same example is significantly smaller than a random sampled baseline, as confirmed by a *paired Wilcoxon signed-rank test*. For the baseline, on each image, we uniformly sample the same number $|\mathcal{S}^{\ell}_m|$ patch tokens at random locations, recompute the distances, and average over repeated draws. This confirms that sink tokens recur in corresponding background regions rather than appearing at arbitrary locations.

**Masking sinks leaves predictions unchanged.** We remove all incoming attention to identified visual sinks by editing the attention mask at inference: for each layer $\ell$ and for every query token $i$, we set $Z^{\ell}_{i,j} = -\infty$ for each key $j \in \mathcal{S}^{\ell}$, which forces $\alpha^{\ell}_{i,j} = 0$ for all queries and all layers. We then compare the model's original outputs with the masked run on the same inputs using the *answer-flip rate*, defined as the fraction of examples whose predicted answer string changes after masking. We report this rate with a *Wilson 95% confidence interval* for binomial proportions; when no flips are observed, we state a conservative 95% upper bound on the true flip probability using the *rule of three*, i.e., $3/n$ for $n$ evaluated items, which closely matches the one-sided Clopper–Pearson bound in this case. Across models, flip rates remain within tight Wilson intervals near zero, indicating that eliminating attention to sink keys does not materially alter predictions. This outcome is consistent with prior reports that attention sinks behave as surplus-attention anchors whose removal has minimal effect on observable outputs (Gu et al., 2025; Kang et al., 2025).

**Positional skew toward earlier images.** Within the same multi-image example, sink strength is not uniform across images. We quantify per-image sink strength at layer $\ell$ by the *Sink Attention Share*

$$\zeta^{\ell}_m \;=\; \frac{\sum_{i \in \mathcal{V}_m} \sum_{j \in \mathcal{S}^{\ell}_m} \alpha^{\ell}_{i,j}}{\sum_{i \in \mathcal{V}_m} \sum_{j \in \mathcal{V}_m} \alpha^{\ell}_{i,j}}, \tag{7}$$

where $\alpha^{\ell}_{i,j}$ are attention weights averaged over heads. Intuitively, $\zeta^{\ell}_m$ measures what fraction of the attention budget directed to image $m$ is absorbed by its sink tokens rather than informative visual tokens, so larger values indicate more attention trapped in sinks. As shown in Fig. 2b, earlier images exhibit consistently larger $\zeta_m$ than later ones; a paired Wilcoxon signed-rank test comparing the first and last image within each example rejects the null of equal medians, and a within-example regression of $\zeta_m$ on the image index $m$ yields a negative, statistically significant slope. This pattern aligns with the one-way access imposed by causal masking (Wu et al., 2025): an image that appears earlier is exposed to more downstream queries, and we observe that $\zeta_m$ increases with this exposure; moreover, permuting image order within the same example shifts $\zeta_m$ toward the images moved earlier, reinforcing the link between positional access and sink accumulation.

## 3.2 Measuring Attention Fragmentation

**Entropy-based measurement of image-level fragmentation.** Prior analyses of Transformer attention report layer-dependent patterns suggesting that attention can become more task-specific deeper in the stack (Voita et al., 2019; Abnar & Zuidema, 2020). In multi-image examples, attention allocation may evolve with depth: early layers can distribute mass more broadly, whereas later layers are expected to place relatively more weight on whichever image carries decisive evidence rather than maintain an even spread. For example, a matching-style query such as "Which image shows the red umbrella?", would expect more attention to be allocated to the image with a red umbrella in the late layers rather than spreading evenly across the images. Following previous work that uses entropy to characterize dispersion of attention (Hyeon-Woo et al., 2023; Araabi et al., 2024), we measure fragmentation at the image level using entropy to test whether attention becomes concentrated or remains dispersed in late layers. For decoder layer $\ell$ and query token $i$, aggregate attention to image $m$ by defining $p_m(i,\ell) = \sum_{j \in \mathcal{V}_m} \alpha_{ij}^{\ell}$ and the total visual mass $w(i,\ell) = \sum_{j \in \mathcal{V}} \alpha_{ij}^{\ell}$. We then normalize over visual mass to obtain a distribution across images $\tilde{p}_m(i,\ell) = p_m(i,\ell)/w(i,\ell)$ (defined when $w(i,\ell) > 0$), so that $\sum_{m=1}^{M} \tilde{p}_m(i,\ell) = 1$. The normalized Shannon entropy is

$$H_{\text{norm}}(i,\ell) \;=\; \frac{-\sum_{m=1}^{M} \tilde{p}_m(i,\ell) \log \tilde{p}_m(i,\ell)}{\log M}, \tag{8}$$

which lies in $[0,1]$: values near 0 indicate concentrated focus on one image; values near 1 indicate a uniform spread across images. In our analysis, higher normalized entropy demonstrates attention fragmentation at the image level, whereas lower values indicate concentration of attention. Mechanistically, if each image contains background sink tokens that attract a comparable share of attention, the per-image attention masses $p_m(i,\ell)$ tend toward equality (i.e., $\approx 1/M$), which raises $H_{\text{norm}}$.

**Fragmentation persists throughout layers.** Following prior work that examines attention patterns across all layers (Abnar & Zuidema, 2020; Zhai et al., 2023), we summarize normalized entropy by depth using quartile bins. We use a Dirichlet(1) compositional reference because it is uniform over the $M$-simplex and encodes no preference among images. As shown in Fig. 2c, relative to this reference, early-layer entropy already lies in the upper tail of its Monte Carlo distribution, indicating high dispersion compared to an uninformed spread. Entropy shows no meaningful reduction as depth increases: paired Wilcoxon signed-rank tests between adjacent quartiles are not significant after Holm correction, and per-example Spearman correlations between layer index and entropy center near zero. By the final quartile, the median entropy remains in the upper tail of the Dirichlet(1) reference, underscoring persistent fragmentation rather than concentrating on more relevant images.

**Link between high entropy, skewed sinks, and recency bias.** We found that fragmentation persists under permutation of image order, and answers flip with a recency bias; images placed later in the sequence disproportionately influence predictions, consistent with one-way access under causal masking and mirroring behavior reported in recent position-bias studies (Tian et al., 2025; Wu et al., 2025). Let $p_m(i,\ell) = \sum_{j \in \mathcal{V}_m} \alpha_{i,j}^{\ell}$ denote the per-image attention mass and let $\zeta_m^{\ell}$ be the sink share on image $m$ (Eq. (7)), where $\alpha_{i,j}^{\ell}$ are attention weights averaged over heads. Let $w(i,\ell) = \sum_{m=1}^{M} p_m(i,\ell) = \sum_{j \in \mathcal{V}} \alpha_{i,j}^{\ell}$ be the total visual attention mass. Define the non-sink attention mass $r_m(i,\ell) = p_m(i,\ell)\left(1 - \zeta_m^{\ell}\right)$. If visual attention is evenly distributed across images (high entropy conditional on visual mass), so that $p_m(i,\ell) = w(i,\ell)/M$ for all $m$, then for any images $a,b$, $r_a(i,\ell) - r_b(i,\ell) = \frac{w(i,\ell)}{M}\left(\zeta_b^{\ell} - \zeta_a^{\ell}\right)$, hence $\zeta_a^{\ell} > \zeta_b^{\ell}$ implies $r_a(i,\ell) < r_b(i,\ell)$. More generally, if the masses are near-uniform with $\max_m \left|p_m(i,\ell) - w(i,\ell)/M\right| \le \varepsilon$ for all $m$, then for any $a,b$, $r_a(i,\ell) - r_b(i,\ell) \ge \frac{w(i,\ell)}{M}\left(\zeta_b^{\ell} - \zeta_a^{\ell}\right) - 2\varepsilon$. Here $\varepsilon$ quantifies deviation from uniformity across images; when the normalized entropy $H_{\text{norm}}(i,\ell)$ of the conditional distribution $\tilde{p}_m(i,\ell) = p_m(i,\ell)/w(i,\ell)$ is high, $\varepsilon$ can be bounded via Pinsker's inequality as $\varepsilon \le w(i,\ell)\sqrt{2 \log M \left(1 - H_{\text{norm}}(i,\ell)\right)}$ (natural logs). Because causal masking increases sink share for earlier images, these relations reduce their non-sink mass and bias decisions toward later images, manifesting as recency bias.

## 3.3 Limitations of Post-Softmax Redistribution

**Post-softmax redistribution from prior work.** Prior research on attention sinks in single-image LMMs proposes reallocating the attention mass removed from identified sink tokens to visual non-

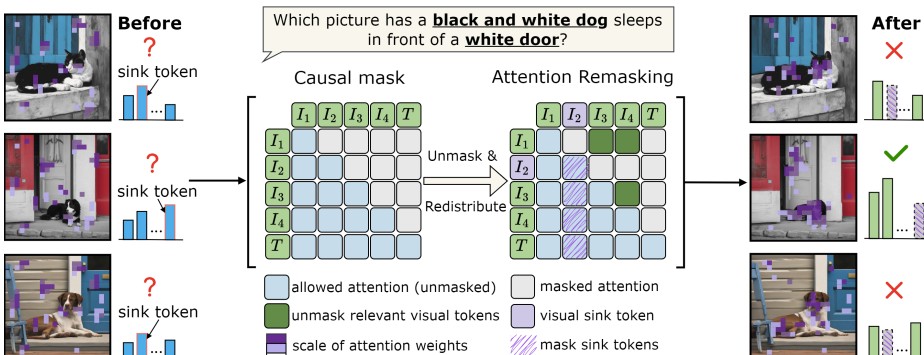

Figure 3: **Attention Remasking.** A post-training edit to pre-softmax attention that masks visual sink tokens and selectively unmasks links between visual tokens guided by task relevance. AR reduces attention fragmentation and mitigates order sensitivity in multi-image LMMs.

sink tokens by directly editing the weights $\alpha_{i,j}^\ell$ in equation 2 computed after the causal mask $\mathcal{M}_{\text{causal}}$ in equation 1 (Kang et al., 2025). The reallocation is proportional to existing non-sink token attention weights, so the image-level distribution $p_m(i, \ell) = \sum_{j \in \mathcal{V}_m} \alpha_{i,j}^\ell$ inherits the baseline pattern; when non-sink token attention is already fragmented across images and skewed toward earlier ones, the same dispersion and skew are preserved, sometimes reinforced by larger sink budgets on earlier images. Because the mask is not altered, forward links from earlier queries to later images cannot be created, so order sensitivity induced by one-way access remains.

**Fragmentation is unchanged after proportional redistribution.** We measure image-level dispersion with the normalized entropy $H_{\text{norm}}(i, \ell)$ in equation 8. Let $H_{\text{norm}}^{\text{base}}$ and $H_{\text{norm}}^{\text{post}}$ denote the scores before and after redistribution. Paired Wilcoxon signed-rank tests find no meaningful difference between $H_{\text{norm}}^{\text{post}}$ and $H_{\text{norm}}^{\text{base}}$, and Hodges–Lehmann estimates of the median change are near zero. Quartile summaries by depth show that the final-quartile median entropy remains in the upper tail of the Dirichlet(1) compositional null both before and after redistribution, indicating that high dispersion persists and late-layer concentration does not emerge.

**Order sensitivity and recency bias persist.** Using the same answer-flip rate and permutation protocol introduced earlier, we observe similar flip frequencies before and after redistribution, indicating that post-softmax reweighting does not stabilize predictions under image reordering. As shown in Fig. 2d, position-wise accuracy continues to increase with later positions, consistent with a recency bias arising from one-way access under causal masking.

## 4 ATTENTION REMASKING

We introduce Attention Remasking (AR) (Fig. 3), a post-training edit to the attention score matrix that targets the multi-image failure induced by attention fragmentation. The objective is twofold: (i) reclaim attention currently assigned to visual sink tokens and (ii) counteract the biased attention distribution left by attention fragmentation, so that attention can flow along task-relevant cross-image links. Let $Z^\ell$ and $\alpha^\ell$ be the pre-softmax scores and row-normalized attention weights defined in equation 1 and equation 2, respectively. Sink tokens are identified as in equation 3 and equation 4, yielding the per-layer set $\mathcal{S}^\ell \subset \{1, \ldots, N\}$. AR edits only the visual-visual submatrix of $Z^\ell$, preserving text autoregression enforced by the causal mask $\mathcal{M}_{\text{causal}}$ in equation 1. AR masks sink tokens by setting $\tilde{Z}_{i,j}^\ell = -\infty$ for all $i$ and all $j \in \mathcal{S}^\ell$, which implements a column-wise block on sinks and removes their incoming attention everywhere. Under causal masking, later images are not accessible as keys to earlier queries, which limits cross-image integration even when the task requires linking evidence across images. AR therefore relaxes the visual parts of $\mathcal{M}_{\text{causal}}$ to allow forward attention from earlier queries to later-image visual tokens when there is semantic evidence that such links are relevant to the task instruction. Building on evidence that patch-level CLIP–text alignment is informative for grounding text queries to image regions(Zhou et al., 2023), we construct a grounded, patch-level relevance distribution over newly unmasked visual keys using CLIP's encoders. First, we locate task-relevant regions with a phrase-grounding detector (e.g., Grounding

Table 1: Average accuracy (%) on five multi-image benchmarks. Full per-task tables and additional models are in the appendix.

| Method | Benchmark (Acc %) | | | | |
|---|---|---|---|---|---|
| | MMIU | MuirBench | MIRB | LIBench | MIBench |
| LLaVA-Interleave-7B | 32.9 | 30.3 | 33.9 | 52.5 | 51.8 |
| + VAR | 34.2 | 32.1 | 34.3 | 53.5 | 52.3 |
| + SoFA | 34.7 | 33.0 | 35.0 | 53.6 | 52.8 |
| + Ours | 37.3 | 35.2 | 36.4 | 56.3 | 55.5 |
| Qwen2-VL-7B | 27.3 | 39.3 | 31.1 | 31.2 | 38.5 |
| + VAR | 28.1 | 40.9 | 32.2 | 32.1 | 39.6 |
| + SoFA | 28.8 | 41.7 | 32.8 | 33.0 | 40.1 |
| + Ours | 31.9 | 44.2 | 34.7 | 35.5 | 42.5 |
| Idefics2-8B | 30.4 | 26.8 | 33.0 | 37.0 | 45.5 |
| + VAR | 30.3 | 27.3 | 33.7 | 37.1 | 45.5 |
| + SoFA | 31.0 | 27.8 | 33.9 | 37.7 | 46.5 |
| + Ours | 34.1 | 30.2 | 35.8 | 41.3 | 50.2 |
| Mantis-SigCLIP-8B | 44.3 | 33.3 | 36.1 | 38.3 | 43.7 |
| + VAR | 45.4 | 33.6 | 37.2 | 39.2 | 44.0 |
| + SoFA | 46.0 | 34.2 | 37.4 | 39.6 | 44.4 |
| + Ours | 47.1 | 36.3 | 38.9 | 42.4 | 47.1 |

DINO), mapping detected boxes to visual-token indices by including all tokens overlapping with the box; denote this grounded candidate set for query $i$ at layer $\ell$ by $\mathcal{U}_i^\ell$. These keys lie in grounded regions and are newly unmasked within the visual–visual block. Let $t \in \mathbb{R}^d$ be the instruction embedding from the CLIP text encoder and $v_j \in \mathbb{R}^d$ the patch/visual-token embedding from the CLIP visual encoder. We compute cosine similarities and normalize to obtain a distribution over $\mathcal{U}_i^\ell$:

$$s_{i,j} = \frac{\langle t, v_j \rangle}{\|t\| \, \|v_j\|}, \qquad \pi_{i,j} = \frac{\exp(s_{i,j})}{\sum_{k \in \mathcal{U}_i^\ell} \exp(s_{i,k})} \quad (j \in \mathcal{U}_i^\ell), \qquad (9)$$

with $\pi_{i,j} = 0$ for $j \notin \mathcal{U}_i^\ell$. This yields a sparse, instruction-conditioned prior that concentrates on grounded, high-similarity patches. We reassign attention released from sinks, guided by $\pi$. For query $i$, let the sink attention at layer $\ell$ be $\eta_i^\ell = \sum_{j \in \mathcal{S}^\ell} \alpha_{i,j}^\ell$. After masking sinks, a plain softmax would redistribute this budget implicitly among remaining keys. Instead, AR explicitly routes the same share to the newly opened, relevant links using $\pi$. Define the target row distribution $\hat{\alpha}_{i,\cdot}^\ell$ by

$$\hat{\alpha}_{i,j}^\ell = (1 - \eta_i^\ell) \frac{\alpha_{i,j}^\ell}{\sum_{k \notin \mathcal{S}^\ell \cup \mathcal{U}_i^\ell} \alpha_{i,k}^\ell} \mathbf{1}\big[j \notin \mathcal{S}^\ell \cup \mathcal{U}_i^\ell\big] + \eta_i^\ell \, \pi_{i,j} \, \mathbf{1}\big[j \in \mathcal{U}_i^\ell\big]. \qquad (10)$$

We realize $\hat{\alpha}_{i,\cdot}^\ell$ by writing scores $\tilde{Z}_{i,j}^\ell = \log \hat{\alpha}_{i,j}^\ell + c_i^\ell$, for all non-$-\infty$ entries, where $c_i^\ell$ is any row-constant canceled by the softmax, while enforcing $-\infty$ on sink columns and on visual links that remain masked. If $\mathcal{S}^\ell = \varnothing$ or $\mathcal{U}_i^\ell = \varnothing$, AR reduces to the identity on row $i$.

## 5 EXPERIMENTS

We evaluate Attention Remasking (AR) on interleaved-token LMMs to test whether it improves multi-image accuracy, reduces attention fragmentation, including lower late-layer entropy and reduced sensitivity to image order, and remains robust under ablations and controls.

### 5.1 EXPERIMENTAL SETTINGS

**Models.** We evaluate a diverse range of open-source multi-image LMMs, including *LLaVA-Interleave-7B* (Li et al., 2025), *Qwen2-VL-7B* (Team, 2025), *Idefics2-8B* (Laurençon et al., 2024), and *Mantis-SigCLIP-8B* (Jiang et al., 2024). For completeness, Appendix C expands to the full set of multi-image LMMs covered by recent benchmarks.

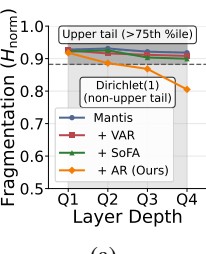 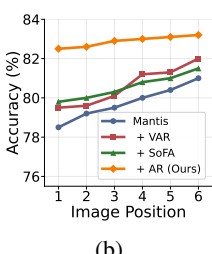 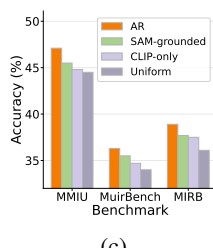 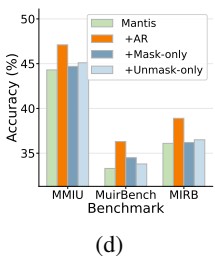

      (a)              (b)              (c)              (d)

Figure 4: **Attention Remasking (AR): results and ablations.** (a) *Fragmentation vs. depth:* AR lowers normalized entropy compared with post-training baselines, indicating reduced attention fragmentation. (b) *Order sensitivity:* AR reduces position-wise accuracy skew relative to baselines. (c) *Ablation on relevance scores:* accuracy when unmasking uses different patch-level scores. (d) *Ablation on masking strategy:* mask-only and unmask-only variants underperform full AR.

**Baselines.** We compare against two training-free post-hoc methods related to attention fragmentation. SoFt Attention (*SoFA*) linearly interpolates the attention between the standard causal attention and the bidirectional attention without masking, opening links from earlier queries to later images (Tian et al., 2025). Visual Attention Redistribution (*VAR*) moves post-softmax attention mass from identified sink tokens to visual non-sink tokens (Kang et al., 2025).

**Tasks and benchmarks.** We adopt a comprehensive multi-image evaluation covering a wide range of complementary tasks. *MMIU* (Meng et al., 2025) covers seven relationship types with tasks that test comparison, retrieval, and spatial/temporal integration; *MuirBench* (Wang et al., 2025) spans twelve tasks built around relation categories such as multiview, ordering, and temporal reasoning; *MIRB* (Zhao et al., 2024) groups tasks into perception, visual world knowledge, reasoning, and multi-hop reasoning; we also use the multi-image subset of *MIBench* (Liu et al., 2024a), which focuses on five tasks of reasoning and comparison, and subset of LLaVA-Interleave Bench (Li et al., 2025), spanning nine tasks on multi-image reasoning, which we refer to as *LIBench*.

**Implementation details.** We use each model's official checkpoints and default prompting templates, and we run evaluation with greedy decoding ($temperature = 0$) to remove sampling variance. For multiple-choice question items, we adopt the benchmark-provided answer extraction and normalization scripts for scoring. We locate instruction-relevant regions with GroundingDINO-SwinT-OGC with default settings. All experiments are conducted on eight A100 GPUs.

## 5.2 MAIN RESULTS

**Overall accuracy.** Across all five multi-image benchmarks and every evaluated open-source LMMs, AR improves average accuracy over the base model. Table 1 reports benchmark-level means; full per-task, per-model results appear in the Appendix C.

**Level of fragmentation.** Fig. 4a plots the normalized entropy $H_{norm}$ by layer-depth quartiles. Baselines lie in the upper tail of a Dirichlet(1) compositional null across depth, indicating dispersed allocation over images. AR shifts the curve downward at all depths, with a statistically significant reduction in the final quartile as confirmed by paired Wilcoxon signed-rank tests with Holm correction. Fig. 4b shows accuracy as a function of image position under controlled permutations. Baselines display a clear positive slope, evidencing recency bias. AR both raises the curve and flattens the slope; permutation flip rates drop, with Wilson intervals remaining strictly below the baseline. For comparison baselines, SoFA reduces the position–accuracy slope while late-layer entropy remains high, and VAR leaves entropy unchanged and the position–accuracy slope largely intact, reflecting that proportional post-softmax reweighting preserves the pre-existing fragmented pattern.

**Discussion.** SoFA interpolates toward unmasked attention using logits that were never trained under the causal mask, opening all links without preference for task-relevant ones; this weakly counters positional bias effects but does not consolidate attention, as fragmentation remains high. VAR moves probability mass only within the already-masked distribution and in proportion to current non-sink weights; it cannot create forward links and therefore inherits both dispersion and positional skew. AR differs by masking sink keys, explicitly unmasking a sparse set of instruction-relevant cross-

image links, and reallocating the freed attention to those links, which jointly reduces fragmentation and order sensitivity while improving task accuracy.

## 5.3 Ablation Study

**Relevance score.** We compare the *DINO-grounded CLIP* score used in AR with three alternatives. *SAM-grounded CLIP* replaces the detector with SAM (Kirillov et al., 2023): images are segmented, segments are ranked by the average CLIP text–patch similarity, the top segments are retained, and the resulting token-wise scores are normalized over the newly unmasked keys. *CLIP-only patches* remove gating and use raw CLIP text–patch cosine similarities for all visual tokens. *Uniform* distributes the freed attention mass evenly across all newly unmasked tokens. Fig. 4c shows that both grounded CLIP scores attain the largest accuracy. Uniform reallocation provides the weakest improvement and often leaves fragmentation and order sensitivity largely unchanged.

**Masking and unmasking.** We compare two ablations against the full AR edit. *Mask-only* removes all incoming attention to identified sinks by setting their columns to $-\infty$ and then renormalizing each row over the remaining keys. This diffuses the budget that was trapped in sinks across the pre-existing non-sink pattern, so image-level dispersion and order sensitivity largely persist; accordingly. *Unmask-only* relaxes the visual mask to open cross-image links and assigns scores equivalent to scores in sinks to those links in proportion to the grounded, patch-level CLIP relevance, but leaves the rest of each row unchanged. Fig. 4d shows that for both variants, accuracy remains close to the baseline. The full AR masks sink and reallocates the freed budget to the newly unmasked, instruction-relevant links, and achieves the largest accuracy improvements among the variants.

## 6 Related Work

**Multi-image understanding.** Recent large multimodal models accept multiple images by incorporating visual tokens with texts into a single causal sequence so that all tokens share one attention space (Li et al., 2025; Team, 2025; Jiang et al., 2024; Cai et al., 2024; Lu et al., 2024). To evaluate performance on multi-image tasks, recently proposed benchmarks provide broad coverage with differing taxonomies. For example, MMIU (Meng et al., 2025) spans relation types such as comparison, retrieval, and spatial or temporal integration, while MuirBench (Wang et al., 2025) aggregates diverse relation-centric tasks, and MIRB (Zhao et al., 2024) targets perception, world knowledge, and multi-hop reasoning. Beyond evaluation, SoFA (Tian et al., 2025) mitigates positional bias by interpolating causal with bidirectional attention via a weighting hyperparameter, and Multi-image Augmented Direct Preference Optimization (Liu et al., 2025) augments preference data with multi-image examples by extending single-image data with unrelated images to improve task alignment.

**Attention sinks.** Attention sinks refer to tokens that attract disproportionately high attention despite offering little semantic utility. Recent work shows they emerge during LLM pretraining, concentrate at fixed positions such as the first token or special tokens, and correlate with massive activation in a few hidden dimensions; masking or removing them has minimal effect, suggesting they store surplus attention rather than useful signals (Gu et al., 2025). Attention sinks have also been exploited for efficiency: StreamingLLM (Xiao et al., 2024) retains sink keys to stabilize sliding-window attention and reduce KV cache, while OrthoRank (Shin et al., 2025) uses sink orthogonality to prune tokens. In multimodal settings, sinks cluster on background patches, motivating redistribution of attention to non-sink tokens (Kang et al., 2025). Our work builds on these observations but targets multi-image trained models, where repeated, position-skewed sinks fragment cross-image attention.

## 7 Conclusion

We introduced Attention Remasking (AR), a post-training edit to the attention scores that removes visual sink tokens and reinstates task-relevant cross-image links under causal masking. Our analyses revealed repeated, position-skewed sinks across images, which fragment attention patterns; AR reduces fragmentation and improves accuracy on multi-image benchmarks without retraining or hyperparameter tuning. AR provides a simple, general tool for multi-image understanding in large multimodal models and moves toward stronger cross-image integration. Future work includes analyzing attention dynamics and extending AR to video modeling if similar issues arise.

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

## A    ETHICS STATEMENT

This work does not involve human subjects, personally identifiable information, or sensitive user data, and therefore does not require IRB approval. All experiments are conducted on publicly available multimodal datasets and established evaluation benchmarks. We strictly follow dataset licenses and usage policies, and we do not release or create new data containing private or restricted content. Our proposed method, Attention Remasking (AR), is a post-training intervention aimed at improving the reasoning ability of large multimodal models on multi-image tasks. It does not introduce additional risks beyond those already associated with LMMs. Nevertheless, we acknowledge broader ethical considerations: improved multimodal reasoning may have dual-use implications, such as being applied in surveillance or manipulative content generation. We stress that our contributions are intended for advancing safe and transparent research in multi-image understanding and should be deployed responsibly, with attention to fairness, accountability, and potential downstream misuse. We have no conflicts of interest or external sponsorship that could influence this work.

## B    ADDITIONAL ANALYSIS

**Qualitative Analysis.** We qualitatively analyse the visual attention maps of the base model and AR in Fig. 5. Without AR, the base model distributes attention across sink tokens and irrelevant regions, leading to fragmented focus and an incorrect answer. With AR, attention is redirected to task-relevant regions, restoring cross-image reasoning and yielding the correct answer.

**Head Selection.** VAR applies redistribution only to image-centric heads, in which attention concentrates on visual non-sink tokens (Kang et al., 2025). In multi-image, interleaved LMMs such

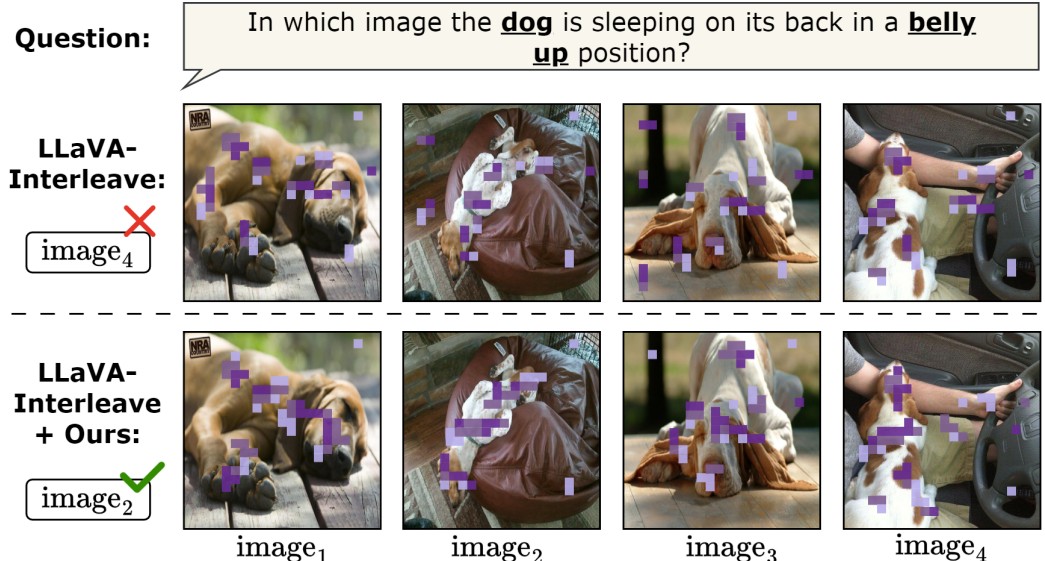

Figure 5: **Qualitative analysis of AR.** Visual attention maps before and after applying AR indicate that, by masking sink tokens and unmasking task-relevant visual attention, AR (bottom) reduces distractions and restores cross-image focus, leading to more accurate responses by seeing the image more effectively.

as LLaVA-Interleave Li et al. (2025); Jiang et al. (2024), the decoder attends over a key set dominated by visual tokens because each image contributes hundreds of ViT/CLIP patch tokens, whereas the prompt contributes only tens of text tokens. Consequently, a large fraction of heads satisfy the image-centric criterion even without explicit selection. Empirically, restricting redistribution to VAR-selected heads yields accuracy that is nearly identical to applying redistribution across all heads, as shown in Fig. 6a.

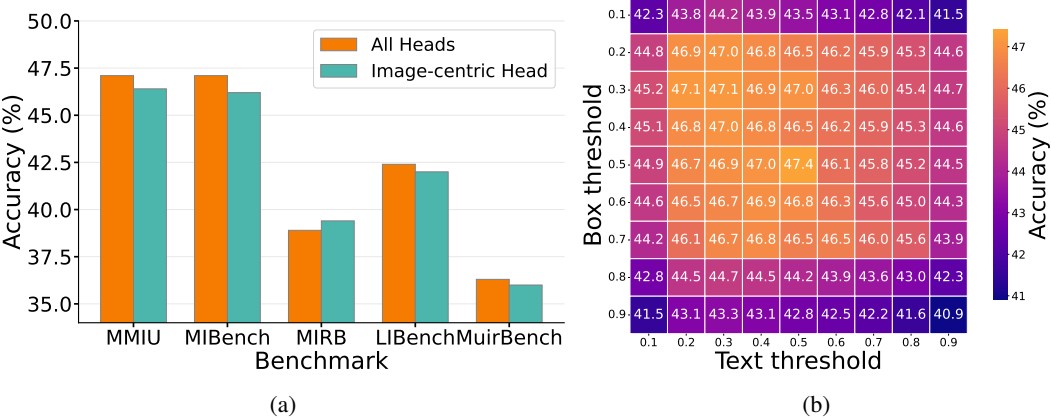

(a)

(b)

Figure 6: **Additional analysis of AR.** (a) *Effect of head selection:* Applying redistribution to all heads or only image-centric heads yields nearly identical accuracy across benchmarks. (b) *Sensitivity analysis of Grounding DINO hyperparameters on MMIU benchmark:* Performance remains stable across reasonable threshold ranges, with degradation only at extreme values.

**Grounding DINO.** Our method relies on Grounding DINO to identify task-relevant regions in images, which are then used to guide attention remasking. Grounding DINO has two hyperparameters: *box_threshold* (minimum confidence for box detection) and *text_threshold* (minimum text-image similarity for grounding). Since we had no validation set available for hyperparameter tuning, we

Table 2: Experimental results on general single-image vision-language task.

| Model | VQA$^{\text{v2}}$ | GQA | VizWiz | SQA$^{\text{I}}$ | VQA$^{\text{T}}$ | MME | MMB$^{\text{en}}$ | SEED$^{\text{I}}$ | LLaVA$^{\text{W}}$ | MM-Vet |
|---|---|---|---|---|---|---|---|---|---|---|
| LLaVA-1.5-7B | 78.5 | 62.0 | 50.0 | 66.8 | 58.2 | 1495.5 | 64.3 | 58.6 | 65.4 | 31.1 |
| + VAR | 78.6 | 63.5 | 53.7 | 67.3 | 58.6 | 1513.8 | 65.1 | 60.7 | 68.1 | 33.7 |
| + Ours | 78.6 | 63.0 | 54.0 | 68.1 | 58.0 | 1510.4 | 64.7 | 59.9 | 67.1 | 34.4 |
| VILA-13B | 80.8 | 63.3 | 60.6 | 73.7 | 66.6 | 1507.1 | 70.3 | 62.8 | 73.0 | 38.8 |
| + VAR | 81.2 | 63.6 | 64.2 | 74.7 | 67.3 | 1512.7 | 71.7 | 63.0 | 75.7 | 39.7 |
| + Ours | 81.6 | 62.4 | 63.4 | 75.1 | 66.9 | 1510.7 | 72.3 | 62.4 | 74.7 | 40.1 |
| Qwen2-VL-7B | 82.5 | 64.5 | 65.4 | 74.1 | 84.3 | 1672.3 | 83.0 | 77.9 | 75.6 | 63.2 |
| + VAR | 82.8 | 64.7 | 67.7 | 74.2 | 84.9 | 1688.5 | 83.3 | 78.1 | 77.3 | 63.5 |
| + Ours | 82.6 | 65.3 | 66.9 | 73.8 | 85.3 | 1690.0 | 84.0 | 78.0 | 76.7 | 64.0 |
| InternVL2-8B | 82.0 | 63.2 | 63.0 | 74.2 | 77.3 | 1648.1 | 81.7 | 76.2 | 73.2 | 60.0 |
| + VAR | 82.5 | 63.5 | 65.1 | 74.7 | 78.0 | 1655.4 | 82.3 | 77.1 | 75.1 | 61.2 |
| + Ours | 82.0 | 64.0 | 64.1 | 75.9 | 78.9 | 1650.9 | 83.6 | 77.5 | 74.6 | 61.8 |

used the default values of `box_threshold=0.35` and `text_threshold=0.25` throughout all our experiments.

To verify that our results are not dependent on these specific hyperparameter choices, we conduct a sensitivity analysis by evaluating all combinations of threshold values from 0.1 to 0.9. Fig. 6b shows the accuracy heatmap across different threshold combinations. The results show that our method is robust across a wide range of reasonable threshold values, performance largely degrades only when either threshold reaches extreme values, confirming that AR is insensitive to the precise choice of these parameters.

**Single Image Performance.** A natural concern is whether our AR method might undermine single-image performance. To verify this, we evaluate AR on single-image vision–language benchmarks, with results reported in Table 2. We find that AR does not harm performance on single-image tasks; it yields consistent improvements across most benchmarks. This effect shows that by masking attention sinks and mitigating the positional bias, AR reallocates attention away from background patches that contribute little to prediction, thereby allowing the model to attend more effectively to task-relevant regions. While the gains are modest, as attention fragmentation is more pronounced in multi-image settings, the improvements show that AR provides a robust enhancement rather than a trade-off.

## C EXPERIMENTAL DETAILS

### C.1 MODEL DETAILS

For completeness, we expand beyond the representative models highlighted in the main text in this section. In addition to LLaVA-Interleave-7B (Li et al., 2025), Qwen2-VL-7B (Team, 2025), Idefics2-8B (Laurençon et al., 2024), and Mantis-SigCLIP-8B (Jiang et al., 2024), we select a diverse set of open-source models, including Mantis-idefics2-8B (Jiang et al., 2024), VILA-2.7B/7B (Lin et al., 2024), Emu2-Chat-37B (Sun et al., 2024b), InternVL2-8B (Chen et al., 2024), InternVL2-Pro (Chen et al., 2024), InternVL1.5-chat (Chen et al., 2024), Mini-InternVL-1.5-2B/4B (Gao et al., 2024), Idefics-9B-Instruct (Bai et al., 2023), DeepSeek-VL-1.3B/7B (Lu et al., 2024), XComposer2-1.8B/7B (Dong et al., 2024), OpenFlamingo-v2 (Awadalla et al., 2023), Qwen-chat (Bai et al., 2023), and Qwen-Base (Bai et al., 2023).

### C.2 REPRODUCIBILITY STATEMENT

**Experimental Settings.** All experiments and evaluations are conducted on eight NVIDIA A100 GPUs. Only the inference step of LMMs is used, without any training.

**Multi-image benchmarks.** We evaluate AR on five recently proposed benchmarks designed to probe multi-image reasoning:

- **MMIU** (Multimodal Multi-Image Understanding) (Meng et al., 2025), is a comprehensive benchmark designed to evaluate multi-image reasoning across seven relationship types: discrete, continuous, low-level, high-level subject, high-level object, 2D, and 3D under-

standing. The benchmark contains over 7,000 carefully curated examples spanning comparison tasks, retrieval scenarios, and spatial-temporal reasoning. Questions are formatted as multiple-choice with 4 options, testing the model's ability to integrate information across multiple images for complex reasoning tasks.

- **MuirBench** (Wang et al., 2025), focuses on robust multi-image understanding across twelve diverse task categories, including counting, action recognition, grounding, matching, ordering, scene understanding, difference detection, cartoon analysis, diagram interpretation, geographic reasoning, attribute recognition, and retrieval. The benchmark contains approximately 2,600 examples designed to test models' resilience to various visual and semantic challenges.

- **MIRB** (Multi-Image Reasoning Benchmark) (Zhao et al., 2024), organizes evaluation around four core competency areas: perception (basic visual recognition across images), visual world knowledge (applying real-world knowledge to visual scenes), reasoning (logical inference and deduction), and multi-hop reasoning (complex reasoning chains spanning multiple images).

- **LIBench** (Li et al., 2025), containing nine task categories: Spot the Difference (SD), Image Edit Instruction (IE), Visual Story Telling (VST), Text-rich VQA (TRVQA), Multi-image VQA (MIVQA), Puzzle solving, Q-Bench quality assessment (QB), ScanQA document understanding (SQ), MathVerse mathematical reasoning (Math), SciVerse scientific reasoning (Sci), Mantis instruction following, BLINK perception tasks, and MMMU multi-discipline understanding, includes approximately 1,500 multi-image examples.

- **MIBench** (Liu et al., 2024a), provides evaluation across two main categories: Multi-Image Instruction following and Multimodal Knowledge-Seeking. The instruction following category includes tasks such as difference comparison (GC), spot the difference (SD), visual reasoning (VR), text-rich understanding (TR), and logical reasoning (LR). The knowledge-seeking category encompasses fine-grained visual recognition (FVR), text-rich image understanding (TRI), visual text knowledge (VTK), and text-visual knowledge (TVK). MIBench contains approximately 2,000 examples.

**Single-image benchmarks.** To verify that AR does not harm performance on standard vision–language tasks, we also report results on ten single-image datasets:

- $VQA^{v2}$ (Goyal et al., 2017): Visual question answering v2.0. A large-scale dataset for visual question answering on natural images, containing open-ended questions about objects, attributes, and relationships.

- GQA (Hudson & Manning, 2019): Question answering on image scene graphs.

- VizWiz (Gurari et al., 2018): We used val set splits for the evaluation. A dataset collected from blind users who photographed their environment and asked related questions.

- $SQA^I$ (Lu et al., 2022): ScienceQA is a dataset collected from elementary and high school science curricula, consisting of 21,208 multimodal multiple-choice science questions. Out of these, 10,332 questions include image context, 10,220 include text context, and 6,532 include both. Most questions are annotated with lectures (17,795) and detailed explanations (19,184) to provide general knowledge and specific reasoning for the correct answers. The dataset spans three subjects: natural science, language science, and social science, and is organized into 26 topics, 127 categories, and 379 skills.

- $VQA^T$ (Singh et al., 2019): TextVQA focuses on visual question answering, where reading embedded text in images (OCR) is essential.

- MME (Yin et al., 2023): A comprehensive evaluation benchmark for multimodal models, covering 14 tasks including object existence, counting, spatial reasoning, OCR, commonsense reasoning, translation, and numerical problem solving. It measures both perception and reasoning abilities.

- $MMB^{en}$ (Liu et al., 2023b): The English subset of MMBench, consisting of ∼3k multiple-choice questions that assess 20 ability dimensions (e.g., perception, reasoning, commonsense). Evaluation adopts the official GPT-4-based scoring pipeline to ensure consistency.

- $SEED^I$ (Li et al., 2023): The image subset of SEED-Bench, targeting visual reasoning with a focus on spatial relationships between objects, a known challenge for current LMMs.

- LLaVA$^{\text{W}}$ (Liu et al., 2023a): LLaVA-Bench (In-the-Wild) contains 24 real-world images with 60 diverse questions spanning domains such as indoor scenes, outdoor environments, memes, paintings, and sketches, testing generalization to complex, unfamiliar visual contexts.

- MM-Vet (Yu et al., 2023): A benchmark designed to assess multimodal models in open-ended visual conversations, including 200 images and 218 questions. Responses are scored by GPT-4 for both accuracy and helpfulness, providing a holistic evaluation of model utility.

## C.3 DETAILED RESULTS

In this section, we report the full per-task and per-model results across all benchmarks.

Table 3: Experiment results on MMIU (Overall (test) to Low-Level).

| Model | Overall (test) | Overall | Discrete | Continuous | Low-level |
|---|---|---|---|---|---|
| Frequency | 30.9 | 31.5 | 29.5 | 29.5 | 38.1 |
| Random | 27.1 | 27.4 | 22.1 | 25.4 | 33.7 |
| *Closed-source LMMs* | | | | | |
| GPT-4o (OpenAI, 2023) | 55.6 | 55.5 | 58.2 | 53.7 | 84.0 |
| Claude3.5 (Anthropic, 2023) | 54.3 | 53.4 | 55.3 | 47.9 | 77.2 |
| Gemini1.5 (Team et al., 2023) | 54.5 | 53.4 | 54.2 | 50.1 | 76.1 |
| Gemini1.0 (Team et al., 2023) | 41.2 | 40.2 | 45.8 | 49.8 | 48.7 |
| *Multi-Image input LMMs* | | | | | |
| Mantis-idefics2-8B (Jiang et al., 2024) | 45.3 | 45.6 | 37.3 | 43.4 | 58.4 |
| + VAR | 46.1 | 46.0 | 38.4 | 44.5 | 59.6 |
| + SoFA | 46.7 | 47.1 | 38.9 | 45.7 | 59.8 |
| + Ours | 48.9 | 49.6 | 41.0 | 47.0 | 61.9 |
| Mantis-SigCLIP-8B (Jiang et al., 2024) | 41.8 | 42.6 | 37.2 | 39.3 | 69.5 |
| + VAR | 42.7 | 43.6 | 38.0 | 40.3 | 71.0 |
| + SoFA | 43.2 | 44.0 | 38.6 | 40.8 | 71.1 |
| + Ours | 45.7 | 46.3 | 40.5 | 43.2 | 73.2 |
| LLaVA-Interleave-7B (Li et al., 2025) | 33.5 | 32.4 | 35.3 | 30.7 | 33.7 |
| + VAR | 34.0 | 33.9 | 35.4 | 31.0 | 34.5 |
| + SoFA | 34.9 | 34.4 | 36.7 | 32.6 | 34.9 |
| + Ours | 37.2 | 36.1 | 39.1 | 34.8 | 38.4 |
| InternVL2-Pro (Chen et al., 2024) | 49.8 | 50.3 | 53.8 | 46.3 | 72.7 |
| + VAR | 51.0 | 50.7 | 54.7 | 47.9 | 71.6 |
| + SoFA | 51.5 | 52.1 | 55.0 | 48.1 | 73.1 |
| + Ours | 53.6 | 54.3 | 57.4 | 50.6 | 76.2 |
| InternVL1.5-chat (Chen et al., 2024) | 38.7 | 37.4 | 43.6 | 46.4 | 42.9 |
| + VAR | 38.8 | 38.7 | 44.7 | 47.8 | 44.1 |
| + SoFA | 38.4 | 39.2 | 45.1 | 48.0 | 44.6 |
| + Ours | 42.7 | 41.3 | 47.0 | 49.9 | 46.8 |
| InternVL2-8B (Chen et al., 2024) | 34.0 | 34.8 | 34.2 | 43.4 | 36.7 |
| + VAR | 35.0 | 35.9 | 35.2 | 44.7 | 37.6 |
| + SoFA | 35.0 | 36.2 | 35.6 | 44.9 | 37.9 |
| + Ours | 38.8 | 38.3 | 37.9 | 46.8 | 40.5 |
| Mini-InternVL-1.5-4B (Gao et al., 2024) | 32.5 | 32.1 | 30.6 | 42.2 | 35.4 |
| + VAR | 33.5 | 33.0 | 31.7 | 43.3 | 36.5 |
| + SoFA | 33.2 | 33.6 | 32.1 | 43.6 | 36.8 |
| + Ours | 36.1 | 36.5 | 34.4 | 45.8 | 39.6 |
| Mini-InternVL-1.5-2B (Gao et al., 2024) | 31.8 | 30.5 | 33.1 | 38.6 | 30.9 |
| + VAR | 32.0 | 31.5 | 34.0 | 39.7 | 31.9 |
| + SoFA | 33.0 | 31.9 | 34.8 | 41.1 | 32.3 |

Table 3 – continued from previous page

| Model | Overall (test) | Overall | Discrete | Continuous | Low-level |
|---|---|---|---|---|---|
| + Ours | 35.6 | 33.8 | 36.7 | 43.9 | 34.8 |
| idefics2-8B (Laurençon et al., 2024) | 27.2 | 27.8 | 22.9 | 19.3 | 42.4 |
| + VAR | 27.6 | 29.1 | 24.0 | 19.7 | 43.5 |
| + SoFA | 28.9 | 29.5 | 24.0 | 21.1 | 43.9 |
| + Ours | 32.5 | 32.4 | 27.1 | 24.5 | 46.8 |
| Idefics-9B-Instruct (Bai et al., 2023) | 13.2 | 12.8 | 23.6 | 7.2 | 11.6 |
| + VAR | 14.3 | 12.7 | 24.7 | 8.1 | 12.1 |
| + SoFA | 14.0 | 14.2 | 25.1 | 8.9 | 12.9 |
| + Ours | 17.6 | 17.2 | 28.0 | 11.7 | 16.8 |
| DeepSeek-VL-7B (Lu et al., 2024) | 24.6 | 24.6 | 16.4 | 10.3 | 39.1 |
| + VAR | 25.9 | 23.4 | 17.1 | 10.8 | 40.0 |
| + SoFA | 27.1 | 25.8 | 18.2 | 12.1 | 40.8 |
| + Ours | 28.8 | 28.0 | 20.6 | 14.4 | 43.9 |
| DeepSeek-VL-1.3B (Lu et al., 2024) | 23.8 | 23.2 | 14.6 | 9.2 | 33.3 |
| + VAR | 24.9 | 24.4 | 15.0 | 10.4 | 34.0 |
| + SoFA | 24.2 | 24.8 | 16.2 | 10.1 | 33.6 |
| + Ours | 27.6 | 26.5 | 18.3 | 13.1 | 36.5 |
| XComposer2-7B (Dong et al., 2024) | 23.4 | 23.5 | 31.9 | 31.6 | 23.4 |
| + VAR | 24.0 | 23.4 | 32.1 | 32.9 | 24.8 |
| + SoFA | 25.1 | 24.9 | 33.5 | 33.0 | 25.2 |
| + Ours | 27.2 | 27.0 | 35.8 | 35.2 | 27.7 |
| XComposer2-1.8B (Dong et al., 2024) | 22.0 | 21.9 | 29.4 | 32.9 | 22.5 |
| + VAR | 23.3 | 22.0 | 30.6 | 34.2 | 23.0 |
| + SoFA | 23.6 | 23.4 | 30.1 | 35.5 | 23.9 |
| + Ours | 26.1 | 25.6 | 33.2 | 38.7 | 26.0 |
| OpenFlamingo-v2 (Awadalla et al., 2023) | 22.7 | 22.3 | 20.8 | 19.5 | 29.6 |
| + VAR | 24.8 | 23.4 | 22.1 | 20.7 | 30.7 |
| + SoFA | 24.2 | 23.8 | 23.6 | 22.2 | 31.2 |
| + Ours | 26.6 | 25.9 | 24.9 | 24.6 | 33.8 |
| Qwen2-VL (Team, 2025) | 33.0 | 27.8 | 25.7 | 27.3 | 28.0 |
| + VAR | 35.0 | 28.7 | 27.8 | 28.1 | 28.5 |
| + SoFA | 32.7 | 28.6 | 27.1 | 28.8 | 30.8 |
| + Ours | 39.9 | 29.9 | 30.2 | 31.9 | 34.1 |
| Qwen-chat (Bai et al., 2023) | 18.0 | 15.9 | 14.7 | 19.5 | 22.3 |
| + VAR | 19.1 | 16.0 | 15.9 | 20.7 | 23.0 |
| + SoFA | 19.6 | 17.5 | 16.0 | 21.8 | 23.1 |
| + Ours | 22.4 | 21.2 | 18.9 | 23.0 | 26.7 |
| Qwen-Base (Bai et al., 2023) | 4.8 | 5.2 | 13.2 | 2.6 | 5.3 |
| + VAR | 6.1 | 6.5 | 14.7 | 3.7 | 6.6 |
| + SoFA | 6.9 | 7.8 | 15.0 | 3.5 | 6.9 |
| + Ours | 10.1 | 10.8 | 18.0 | 8.4 | 11.6 |

Table 3 - Experiment results on MMIU (High-level-sub to Three-D)

| Model | High-level-sub | High-level-obj | Two-D | Three-D |
|---|---|---|---|---|
| Frequency | 29.6 | 36.7 | 27.8 | 30.2 |
| Random | 20.7 | 32.8 | 24.3 | 28.4 |
| *Closed-source LMMs* | | | | |
| GPT-4o (OpenAI, 2023) | 69.2 | 57.5 | 41.7 | 55.4 |
| Claude3.5 (Anthropic, 2023) | 64.8 | 64.5 | 41.9 | 45.1 |

Table 3 – continued from previous page

| Model | High-level-sub | High-level-obj | Two-D | Three-D |
|---|---|---|---|---|
| Gemini1.5 (Team et al., 2023) | 63.9 | 64.9 | 43.3 | 43.0 |
| Gemini1.0 (Team et al., 2023) | 57.9 | 36.7 | 29.7 | 36.7 |
| *Multi-Image input LMMs* | | | | |
| Mantis-idefics2-8B (Jiang et al., 2024) | 54.8 | 56.4 | 37.8 | 40.4 |
| + VAR | 56.1 | 57.6 | 38.1 | 41.2 |
| + SoFA | 57.0 | 57.0 | 39.4 | 42.0 |
| + Ours | 59.0 | 60.3 | 41.8 | 44.7 |
| Mantis-SigCLIP-8B (Jiang et al., 2024) | 46.2 | 52.9 | 30.2 | 40.2 |
| + VAR | 47.6 | 54.3 | 31.2 | 41.7 |
| + SoFA | 47.8 | 54.4 | 32.6 | 42.8 |
| + Ours | 50.1 | 56.9 | 34.8 | 44.2 |
| LLaVA-Interleave-7B (Li et al., 2025) | 35.7 | 33.3 | 34.7 | 27.4 |
| + VAR | 34.9 | 34.5 | 35.6 | 28.7 |
| + SoFA | 37.1 | 35.9 | 36.0 | 29.8 |
| + Ours | 40.6 | 39.2 | 38.4 | 33.1 |
| InternVL2-Pro (Chen et al., 2024) | 70.6 | 58.5 | 38.1 | 42.1 |
| + VAR | 71.8 | 59.9 | 38.2 | 42.5 |
| + SoFA | 72.0 | 60.9 | 39.5 | 43.8 |
| + Ours | 74.1 | 62.3 | 42.0 | 45.9 |
| InternVL1.5-chat (Chen et al., 2024) | 59.1 | 26.0 | 33.6 | 37.0 |
| + VAR | 60.1 | 27.9 | 32.5 | 38.0 |
| + SoFA | 60.4 | 27.4 | 34.9 | 38.0 |
| + Ours | 62.1 | 30.8 | 36.9 | 40.3 |
| InternVL2-8B (Chen et al., 2024) | 47.3 | 32.1 | 30.0 | 32.2 |
| + VAR | 48.0 | 33.5 | 29.1 | 30.3 |
| + SoFA | 49.1 | 33.5 | 31.5 | 33.7 |
| + Ours | 51.1 | 35.8 | 34.0 | 36.4 |
| Mini-InternVL-1.5-4B (Gao et al., 2024) | 47.2 | 29.2 | 27.2 | 30.5 |
| + VAR | 48.0 | 30.6 | 27.6 | 30.8 |
| + SoFA | 48.6 | 31.7 | 28.9 | 32.1 |
| + Ours | 50.5 | 33.2 | 30.9 | 34.2 |
| Mini-InternVL-1.5-2B (Gao et al., 2024) | 37.6 | 28.7 | 27.4 | 25.7 |
| + VAR | 38.0 | 29.8 | 28.6 | 25.9 |
| + SoFA | 39.2 | 30.3 | 29.7 | 27.3 |
| + Ours | 41.7 | 32.4 | 32.3 | 29.8 |
| idefics2-8B (Laurençon et al., 2024) | 45.2 | 26.8 | 33.4 | 25.7 |
| + VAR | 46.0 | 28.1 | 32.5 | 26.9 |
| + SoFA | 46.8 | 28.0 | 34.9 | 27.2 |
| + Ours | 49.2 | 30.9 | 38.3 | 29.5 |
| Idefics-9B-Instruct (Bai et al., 2023) | 27.0 | 12.3 | 12.2 | 8.7 |
| + VAR | 27.2 | 11.2 | 13.1 | 9.6 |
| + SoFA | 28.2 | 13.6 | 13.7 | 10.0 |
| + Ours | 31.0 | 15.9 | 15.7 | 13.3 |
| DeepSeek-VL-7B (Lu et al., 2024) | 32.3 | 34.2 | 32.9 | 16.7 |
| + VAR | 33.5 | 34.6 | 33.9 | 16.9 |
| + SoFA | 33.7 | 36.0 | 34.2 | 18.1 |
| + Ours | 36.4 | 38.4 | 37.6 | 22.3 |
| DeepSeek-VL-1.3B (Lu et al., 2024) | 24.9 | 30.8 | 32.7 | 19.0 |
| + VAR | 25.1 | 32.2 | 33.8 | 20.3 |
| + SoFA | 26.4 | 32.3 | 34.7 | 20.9 |

Table 3 – continued from previous page

| Model | High-level-sub | High-level-obj | Two-D | Three-D |
|---|---|---|---|---|
| + Ours | 28.7 | 34.9 | 36.6 | 23.9 |
| XComposer2-7B (Dong et al., 2024) | 34.3 | 20.0 | 18.7 | 18.0 |
| + VAR | 34.7 | 21.1 | 19.2 | 19.3 |
| + SoFA | 36.0 | 21.2 | 20.1 | 19.9 |
| + Ours | 38.3 | 23.8 | 22.6 | 22.0 |
| XComposer2-1.8B (Dong et al., 2024) | 36.2 | 15.3 | 20.9 | 14.6 |
| + VAR | 37.0 | 16.0 | 22.0 | 15.0 |
| + SoFA | 38.1 | 17.0 | 22.8 | 16.3 |
| + Ours | 40.3 | 20.2 | 24.7 | 18.6 |
| OpenFlamingo-v2 (Awadalla et al., 2023) | 24.6 | 26.9 | 17.2 | 21.7 |
| + VAR | 25.7 | 26.1 | 18.5 | 22.9 |
| + SoFA | 25.1 | 28.6 | 19.9 | 22.4 |
| + Ours | 28.6 | 31.0 | 21.3 | 25.7 |
| Qwen2-VL (Team, 2025) | 31.0 | 24.8 | 20.7 | 27.0 |
| + VAR | 31.0 | 25.1 | 22.8 | 28.5 |
| + SoFA | 32.1 | 26.6 | 23.1 | 28.8 |
| + Ours | 35.9 | 29.9 | 24.2 | 31.1 |
| Qwen-chat (Bai et al., 2023) | 21.3 | 14.8 | 10.5 | 17.1 |
| + VAR | 22.6 | 16.1 | 10.8 | 18.0 |
| + SoFA | 22.9 | 16.9 | 12.1 | 18.7 |
| + Ours | 25.6 | 18.9 | 13.4 | 20.8 |
| Qwen-Base (Bai et al., 2023) | 10.1 | 4.6 | 2.8 | 3.8 |
| + VAR | 11.5 | 5.7 | 3.1 | 5.1 |
| + SoFA | 11.5 | 6.1 | 4.2 | 5.3 |
| + Ours | 14.6 | 8.4 | 6.7 | 7.9 |

Table 4: Experiment results on MuirBench (Overall to Scene).

| Model | Overall | Count | Action | Ground | Match | Order | Scene |
|---|---|---|---|---|---|---|---|
| Random Choice | 24.0 | 21.0 | 23.4 | 25.0 | 24.1 | 22.8 | 25.0 |
| Human | 93.2 | 94.9 | 97.6 | 85.7 | 94.8 | 87.5 | 94.6 |
| *Closed-source LMMs* | | | | | | | |
| GPT-4o (OpenAI, 2023) | 68.0 | 49.2 | 44.5 | 36.9 | 86.9 | 23.4 | 71.5 |
| GPT-4-Turbo (OpenAI, 2023) | 62.3 | 42.3 | 39.6 | 53.6 | 80.4 | 35.9 | 59.1 |
| Gemini Pro (Team et al., 2023) | 49.4 | 28.6 | 36.0 | 28.6 | 66.6 | 12.5 | 59.1 |
| *Multi-Image input LMMs* | | | | | | | |
| Mantis-8B-Idefics2 (Jiang et al., 2024) | 44.5 | 38.5 | 33.5 | 26.2 | 53.9 | 18.8 | 57.0 |
| + VAR | 45.0 | 39.0 | 33.9 | 27.0 | 54.5 | 18.3 | 57.4 |
| + SoFA | 45.8 | 39.6 | 34.7 | 27.2 | 55.0 | 20.1 | 58.2 |
| + Ours | 48.5 | 42.4 | 36.9 | 30.3 | 57.6 | 24.5 | 60.5 |
| Mantis-8B-clip-llama3 (Jiang et al., 2024) | 37.4 | 29.1 | 36.6 | 21.4 | 43.3 | 18.8 | 57.0 |
| + VAR | 38.1 | 30.0 | 37.3 | 20.6 | 43.8 | 19.9 | 58.0 |
| + SoFA | 39.0 | 30.2 | 38.1 | 22.8 | 45.0 | 20.1 | 58.5 |
| + Ours | 41.2 | 32.4 | 39.9 | 24.7 | 47.9 | 22.0 | 60.2 |
| Mantis-8B-siglip-llama3 (Jiang et al., 2024) | 36.1 | 27.4 | 37.2 | 22.6 | 43.8 | 7.8 | 54.3 |
| + VAR | 37.0 | 28.4 | 37.4 | 23.9 | 44.0 | 8.9 | 55.0 |
| + SoFA | 37.2 | 28.5 | 38.1 | 23.0 | 44.7 | 9.6 | 55.3 |
| + Ours | 39.6 | 30.5 | 40.2 | 25.8 | 46.6 | 11.7 | 57.5 |
| LLaVA-Interleave-7B (Li et al., 2025) | 41.0 | 33.0 | 40.0 | 26.5 | 49.5 | 22.0 | 60.0 |
| + VAR | 42.3 | 34.5 | 41.5 | 28.1 | 51.2 | 23.8 | 61.6 |

Table 4 – continued from previous page

| Model | Overall | Count | Action | Ground | Match | Order | Scene |
|---|---|---|---|---|---|---|---|
| + SoFA | 43.0 | 35.2 | 42.3 | 29.0 | 52.0 | 24.5 | 62.4 |
| + Ours | 45.6 | 37.9 | 44.8 | 31.7 | 54.9 | 27.4 | 64.7 |
| Idefics-9B-Instruct (Laurençon et al., 2023) | 35.4 | 29.9 | 28.1 | 13.1 | 36.0 | 12.5 | 27.4 |
| + VAR | 36.0 | 30.0 | 29.0 | 14.7 | 35.4 | 13.0 | 28.2 |
| + SoFA | 36.9 | 31.2 | 29.5 | 14.3 | 37.7 | 13.8 | 28.6 |
| + Ours | 39.0 | 33.3 | 31.4 | 16.2 | 39.7 | 15.5 | 30.6 |
| Idefics2-8B (Laurençon et al., 2024) | 26.1 | 21.8 | 26.2 | 26.2 | 24.8 | 15.6 | 56.5 |
| + VAR | 27.0 | 23.0 | 27.0 | 26.6 | 26.0 | 16.8 | 57.7 |
| + SoFA | 27.7 | 23.3 | 27.9 | 27.9 | 26.0 | 17.1 | 58.2 |
| + Ours | 30.0 | 26.4 | 29.8 | 31.9 | 28.5 | 18.0 | 60.8 |
| Emu2-Chat-37B (Sun et al., 2024b) | 33.6 | 31.2 | 27.4 | 26.2 | 37.3 | 15.6 | 48.4 |
| + VAR | 34.8 | 32.5 | 28.6 | 27.0 | 38.1 | 16.8 | 49.9 |
| + SoFA | 35.6 | 32.5 | 28.2 | 27.9 | 38.8 | 17.1 | 50.7 |
| + Ours | 36.3 | 35.9 | 30.8 | 29.6 | 41.9 | 19.0 | 52.9 |
| VILA-13B (Lin et al., 2024) | 33.1 | 19.7 | 28.7 | 25.0 | 41.0 | 10.9 | 56.5 |
| + VAR | 34.0 | 21.6 | 29.0 | 25.2 | 42.0 | 12.9 | 57.5 |
| + SoFA | 34.7 | 21.4 | 30.0 | 26.5 | 42.5 | 12.2 | 58.1 |
| + Ours | 36.7 | 23.4 | 32.0 | 28.2 | 44.2 | 14.5 | 60.6 |
| OpenFlamingo-v2-9B (Awadalla et al., 2023) | 23.7 | 21.8 | 26.8 | 31.0 | 24.1 | 21.9 | 22.6 |
| + VAR | 24.4 | 23.0 | 27.1 | 32.7 | 25.0 | 23.0 | 22.8 |
| + SoFA | 25.0 | 23.2 | 28.0 | 32.6 | 25.6 | 23.5 | 24.0 |
| + Ours | 27.1 | 25.1 | 29.9 | 34.3 | 27.7 | 25.1 | 25.9 |
| Qwen2-VL (Team, 2025) | 34.0 | 28.6 | 29.5 | 21.0 | 34.5 | 15.0 | 37.0 |
| + VAR | 35.6 | 30.2 | 31.3 | 22.7 | 36.2 | 16.6 | 38.9 |
| + SoFA | 36.4 | 31.0 | 32.1 | 23.6 | 37.0 | 17.3 | 39.6 |
| + Ours | 38.8 | 33.4 | 34.5 | 25.9 | 39.1 | 19.1 | 41.8 |
| Qwen-VL (Bai et al., 2023) | 30.2 | 25.4 | 27.0 | 17.6 | 31.1 | 12.9 | 34.0 |
| + VAR | 31.3 | 26.5 | 28.1 | 18.7 | 32.0 | 13.7 | 35.2 |
| + SoFA | 32.1 | 26.2 | 28.9 | 20.4 | 32.9 | 13.8 | 36.0 |
| + Ours | 35.6 | 28.7 | 30.4 | 21.3 | 34.1 | 15.4 | 38.1 |
| Qwen-Base (Bai et al., 2023) | 21.5 | 18.0 | 19.3 | 10.2 | 22.1 | 9.0 | 24.2 |
| + VAR | 22.6 | 19.1 | 19.2 | 10.1 | 23.0 | 9.0 | 24.1 |
| + SoFA | 23.4 | 18.8 | 20.1 | 10.9 | 23.1 | 9.5 | 24.9 |
| + Ours | 25.0 | 21.3 | 22.4 | 12.8 | 25.5 | 11.5 | 27.3 |

Table 4 – Experiment results on MuirBench (Difference to Retrieval)

| Model | Diff | Cartoon | Diagram | Geogra | Attribute | Retrieval |
|---|---|---|---|---|---|---|
| Random Choice | 23.2 | 25.0 | 29.6 | 25.0 | 20.0 | 21.3 |
| Human | 92.9 | 82.1 | 99.0 | 98.0 | 87.8 | 86.3 |
| *Closed-source LMMs* | | | | | | |
| GPT-4o (OpenAI, 2023) | 60.3 | 51.3 | 88.7 | 56.0 | 56.1 | 80.1 |
| GPT-4-Turbo (OpenAI, 2023) | 60.6 | 52.6 | 79.2 | 57.0 | 50.5 | 64.0 |
| Gemini Pro (Team et al., 2023) | 45.3 | 47.4 | 64.8 | 48.0 | 41.3 | 43.8 |
| *Multi-Image input LMMs* | | | | | | |
| Mantis-8B-Idefics2 (Jiang et al., 2024) | 28.8 | 38.5 | 67.6 | 26.0 | 48.5 | 35.6 |
| + VAR | 29.9 | 39.0 | 68.5 | 26.0 | 49.7 | 35.8 |
| + SoFA | 30.1 | 39.9 | 69.2 | 27.2 | 50.1 | 37.0 |
| + Ours | 32.4 | 41.9 | 71.5 | 29.3 | 52.3 | 39.5 |
| Mantis-8B-clip-llama3 (Jiang et al., 2024) | 24.1 | 43.6 | 54.3 | 16.0 | 33.7 | 31.9 |

Table 4 – continued from previous page

| Model | Diff | Cartoon | Diagram | Geogra | Attribute | Retrieval |
|---|---|---|---|---|---|---|
| + VAR | 25.3 | 44.8 | 54.8 | 17.0 | 35.0 | 33.0 |
| + SoFA | 25.6 | 45.1 | 55.1 | 17.1 | 35.3 | 33.5 |
| + Ours | 27.4 | 47.1 | 58.0 | 18.9 | 37.2 | 35.6 |
| Mantis-8B-siglip-llama3 (Jiang et al., 2024) | 27.4 | 46.2 | 48.0 | 22.0 | 31.6 | 28.1 |
| + VAR | 28.0 | 46.2 | 49.0 | 23.0 | 32.2 | 28.2 |
| + SoFA | 28.7 | 47.6 | 49.4 | 23.1 | 32.8 | 29.4 |
| + Ours | 30.5 | 49.6 | 51.3 | 25.2 | 34.7 | 31.5 |
| LLaVA-Interleave-7B (Li et al., 2025) | 33.0 | 48.0 | 60.0 | 27.5 | 37.0 | 34.0 |
| + VAR | 34.4 | 49.5 | 61.6 | 28.9 | 38.6 | 35.5 |
| + SoFA | 35.2 | 50.4 | 62.4 | 29.8 | 39.5 | 36.4 |
| + Ours | 37.5 | 52.8 | 64.9 | 32.0 | 41.7 | 38.7 |
| Idefics-9B-Instruct (Laurençon et al., 2023) | 34.4 | 48.7 | 47.0 | 35.0 | 32.7 | 43.5 |
| + VAR | 35.0 | 49.2 | 48.0 | 35.4 | 34.0 | 44.7 |
| + SoFA | 35.9 | 50.0 | 48.4 | 36.1 | 34.3 | 45.1 |
| + Ours | 37.8 | 52.1 | 50.9 | 39.2 | 36.2 | 46.9 |
| Idefics2-8B (Laurençon et al., 2024) | 27.7 | 39.7 | 25.4 | 21.0 | 17.9 | 17.1 |
| + VAR | 28.7 | 39.8 | 25.4 | 21.1 | 18.5 | 18.0 |
| + SoFA | 29.0 | 41.1 | 26.8 | 22.4 | 19.0 | 18.5 |
| + Ours | 33.2 | 43.3 | 28.7 | 26.3 | 20.7 | 20.5 |
| Emu2-Chat-37B (Sun et al., 2024b) | 32.6 | 43.6 | 37.7 | 34.0 | 31.6 | 24.0 |
| + VAR | 33.7 | 44.3 | 38.0 | 35.0 | 32.9 | 24.1 |
| + SoFA | 34.1 | 45.1 | 39.2 | 36.1 | 33.3 | 25.5 |
| + Ours | 36.0 | 47.0 | 41.1 | 37.8 | 35.3 | 27.5 |
| VILA-13B (Lin et al., 2024) | 24.7 | 30.8 | 42.7 | 31.0 | 24.5 | 30.1 |
| + VAR | 25.5 | 30.9 | 42.8 | 32.0 | 25.8 | 30.6 |
| + SoFA | 26.1 | 32.3 | 44.2 | 32.7 | 26.2 | 32.0 |
| + Ours | 28.0 | 34.3 | 47.3 | 34.7 | 28.3 | 34.9 |
| OpenFlamingo-v2-9B (Awadalla et al., 2023) | 21.8 | 25.6 | 31.9 | 25.0 | 18.9 | 15.4 |
| + VAR | 22.7 | 26.1 | 32.5 | 25.1 | 19.3 | 16.7 |
| + SoFA | 23.0 | 27.0 | 33.1 | 26.5 | 20.2 | 17.0 |
| + Ours | 25.1 | 29.3 | 36.2 | 27.6 | 22.4 | 19.1 |
| Qwen2-VL (Team, 2025) | 31.0 | 38.4 | 42.1 | 25.1 | 28.6 | 29.3 |
| + VAR | 32.8 | 40.5 | 44.3 | 26.9 | 30.2 | 31.5 |
| + SoFA | 33.6 | 41.6 | 45.5 | 27.8 | 31.1 | 32.4 |
| + Ours | 35.9 | 44.0 | 47.9 | 29.9 | 33.0 | 34.6 |
| Qwen-VL (Bai et al., 2023) | 29.1 | 36.7 | 40.2 | 23.5 | 26.8 | 27.4 |
| + VAR | 30.1 | 36.8 | 40.0 | 23.4 | 27.0 | 28.0 |
| + SoFA | 29.8 | 37.4 | 40.7 | 24.9 | 27.4 | 28.0 |
| + Ours | 32.2 | 39.6 | 43.0 | 26.3 | 29.4 | 30.0 |
| Qwen-Base (Bai et al., 2023) | 22.0 | 30.2 | 31.5 | 18.0 | 20.1 | 20.9 |
| + VAR | 23.0 | 31.2 | 32.0 | 18.0 | 21.0 | 21.3 |
| + SoFA | 23.4 | 31.5 | 32.7 | 19.1 | 21.3 | 22.0 |
| + Ours | 25.5 | 33.6 | 34.9 | 21.0 | 23.2 | 24.8 |

Table 5: Experiment results on MIRB.

| Model | Reasoning | Knowledge | Perception | Multi-Hop | Average |
|---|---|---|---|---|---|
| Random | 20.8 | 37.6 | 21.4 | 0.0 | 23.0 |
| *Closed-source LMMs* | | | | | |

Table 5 – continued from previous page

| Model | Reasoning | Knowledge | Perception | Multi-Hop | Average |
|---|---|---|---|---|---|
| GPT-4o (OpenAI, 2023) | 81.0 | 55.0 | 53.2 | 40.0 | 57.3 |
| GPT-4V (Achiam et al., 2023) | 75.7 | 50.6 | 49.7 | 36.3 | 53.1 |
| Claude3.5 (Anthropic, 2023) | 70.5 | 47.2 | 45.0 | 33.5 | 49.0 |
| Gemini1.5 (Team et al., 2023) | 66.8 | 44.7 | 43.1 | 31.0 | 46.4 |
| Gemini1.0 (Team et al., 2023) | 55.5 | 35.1 | 34.0 | 24.5 | 37.3 |
| *Multi-Image input LMMs* | | | | | |
| Mantis-idefics2-8B (Jiang et al., 2024) | 44.0 | 30.8 | 36.0 | 0.0 | 27.7 |
| + VAR | 45.4 | 31.2 | 37.2 | 0.0 | 28.5 |
| + SoFA | 45.7 | 31.9 | 37.0 | 0.0 | 28.7 |
| + Ours | 47.8 | 34.0 | 39.1 | 0.0 | 30.2 |
| Mantis-SigCLIP-8B (Jiang et al., 2024) | 60.7 | 38.9 | 44.8 | 0.0 | 36.1 |
| + VAR | 61.1 | 40.7 | 46.9 | 0.0 | 37.2 |
| + SoFA | 62.0 | 40.3 | 47.2 | 0.0 | 37.4 |
| + Ours | 64.2 | 42.0 | 49.3 | 0.0 | 38.9 |
| LLaVA-Interleave-7B (Li et al., 2025) | 56.0 | 36.5 | 43.0 | 0.0 | 33.9 |
| + VAR | 57.1 | 37.0 | 43.0 | 0.0 | 34.3 |
| + SoFA | 58.0 | 37.8 | 44.3 | 0.0 | 35.0 |
| + Ours | 59.2 | 40.4 | 46.0 | 0.0 | 36.4 |
| VILA-2.7B (Lin et al., 2024) | 53.3 | 0.0 | 48.3 | 0.0 | 25.4 |
| + VAR | 54.0 | 31.5 | 49.9 | 0.0 | 33.9 |
| + SoFA | 54.2 | 32.2 | 49.3 | 0.0 | 34.0 |
| + Ours | 56.8 | 34.6 | 53.7 | 0.0 | 36.3 |
| VILA-7B (Lin et al., 2024) | 63.7 | 35.3 | 47.1 | 0.0 | 36.5 |
| + VAR | 64.0 | 34.7 | 47.2 | 0.0 | 36.5 |
| + SoFA | 64.8 | 36.5 | 48.0 | 0.0 | 37.3 |
| + Ours | 67.1 | 38.9 | 50.2 | 0.0 | 39.1 |
| Emu2-Chat-37B (Sun et al., 2024b) | 40.4 | 24.5 | 44.0 | 0.0 | 27.2 |
| + VAR | 40.6 | 25.8 | 44.1 | 0.0 | 27.6 |
| + SoFA | 41.9 | 26.0 | 45.4 | 0.0 | 28.3 |
| + Ours | 44.1 | 28.2 | 47.4 | 0.0 | 29.9 |
| InternVL2-Pro (Chen et al., 2024) | 66.0 | 42.0 | 49.0 | 0.0 | 39.3 |
| + VAR | 67.1 | 42.3 | 50.0 | 0.0 | 39.9 |
| + SoFA | 67.8 | 43.6 | 50.7 | 0.0 | 40.5 |
| + Ours | 70.0 | 45.5 | 52.6 | 0.0 | 42.0 |
| InternVL1.5-chat (Chen et al., 2024) | 52.0 | 30.0 | 42.1 | 0.0 | 31.0 |
| + VAR | 53.0 | 31.4 | 42.3 | 0.0 | 31.7 |
| + SoFA | 53.6 | 31.8 | 43.6 | 0.0 | 32.3 |
| + Ours | 55.8 | 33.7 | 45.5 | 0.0 | 33.8 |
| InternVL2-8B (Chen et al., 2024) | 58.0 | 33.5 | 44.0 | 0.0 | 33.9 |
| + VAR | 57.2 | 34.7 | 44.3 | 0.0 | 34.1 |
| + SoFA | 59.5 | 35.0 | 45.6 | 0.0 | 35.0 |
| + Ours | 61.6 | 37.1 | 47.7 | 0.0 | 36.6 |
| Mini-InternVL-1.5-4B (Gao et al., 2024) | 35.0 | 24.0 | 30.2 | 0.0 | 22.3 |
| + VAR | 36.0 | 24.1 | 30.3 | 0.0 | 22.6 |
| + SoFA | 36.3 | 25.4 | 31.6 | 0.0 | 23.3 |
| + Ours | 38.5 | 27.3 | 33.5 | 0.0 | 24.8 |
| Mini-InternVL-1.5-2B (Gao et al., 2024) | 33.1 | 22.5 | 29.0 | 0.0 | 21.2 |
| + VAR | 34.1 | 22.5 | 30.0 | 0.0 | 21.7 |
| + SoFA | 34.0 | 23.8 | 30.3 | 0.0 | 22.0 |
| + Ours | 36.5 | 25.8 | 32.2 | 0.0 | 23.6 |
| idefics2-8B (Laurençon et al., 2024) | 61.3 | 31.8 | 39.0 | 0.0 | 33.0 |

Table 5 – continued from previous page

| Model | Reasoning | Knowledge | Perception | Multi-Hop | Average |
|---|---|---|---|---|---|
| + VAR | 61.7 | 33.0 | 40.0 | 0.0 | 33.7 |
| + SoFA | 61.9 | 33.3 | 40.6 | 0.0 | 33.9 |
| + Ours | 65.4 | 35.2 | 42.7 | 0.0 | 35.8 |
| Idefics-9B-Instruct (Bai et al., 2023) | 45.9 | 23.5 | 36.9 | 0.0 | 26.6 |
| + VAR | 47.0 | 24.2 | 37.2 | 0.0 | 27.1 |
| + SoFA | 47.4 | 25.1 | 38.3 | 0.0 | 27.7 |
| + Ours | 49.5 | 27.0 | 40.1 | 0.0 | 29.2 |
| DeepSeek-VL-7B (Lu et al., 2024) | 42.0 | 26.0 | 33.0 | 0.0 | 25.3 |
| + VAR | 43.2 | 27.0 | 33.0 | 0.0 | 25.8 |
| + SoFA | 43.3 | 27.6 | 34.3 | 0.0 | 26.3 |
| + Ours | 45.8 | 29.7 | 36.4 | 0.0 | 28.0 |
| DeepSeek-VL-1.3B (Lu et al., 2024) | 30.5 | 20.0 | 27.0 | 0.0 | 19.4 |
| + VAR | 31.4 | 20.1 | 27.8 | 0.0 | 19.8 |
| + SoFA | 31.9 | 21.3 | 28.3 | 0.0 | 20.4 |
| + Ours | 33.8 | 23.1 | 30.1 | 0.0 | 21.8 |
| XComposer2-7B (Dong et al., 2024) | 54.7 | 37.2 | 37.2 | 0.8 | 32.5 |
| + VAR | 55.0 | 37.5 | 38.6 | 1.6 | 33.2 |
| + SoFA | 56.2 | 38.7 | 38.8 | 1.5 | 33.8 |
| + Ours | 58.6 | 41.0 | 41.0 | 3.5 | 36.0 |
| XComposer2-1.8B (Dong et al., 2024) | 46.0 | 29.0 | 34.5 | 0.0 | 27.4 |
| + VAR | 47.1 | 30.0 | 35.7 | 0.0 | 28.2 |
| + SoFA | 47.5 | 30.6 | 36.5 | 0.0 | 28.6 |
| + Ours | 49.7 | 32.7 | 38.1 | 0.0 | 30.1 |
| OpenFlamingo-v2 (Awadalla et al., 2023) | 24.0 | 18.0 | 22.5 | 0.0 | 16.1 |
| + VAR | 25.0 | 19.2 | 23.6 | 0.0 | 16.9 |
| + SoFA | 25.3 | 19.9 | 23.7 | 0.0 | 17.2 |
| + Ours | 27.4 | 21.4 | 25.8 | 0.0 | 18.7 |
| Qwen2-VL (Team, 2025) | 50.2 | 32.6 | 41.5 | 0.0 | 31.1 |
| + VAR | 51.8 | 33.9 | 43.0 | 0.0 | 32.2 |
| + SoFA | 52.7 | 34.5 | 43.8 | 0.0 | 32.8 |
| + Ours | 55.5 | 36.8 | 46.6 | 0.0 | 34.7 |
| Qwen-VL (Bai et al., 2023) | 19.2 | 13.9 | 24.4 | 0.0 | 14.4 |
| + VAR | 19.0 | 14.5 | 25.1 | 0.0 | 14.7 |
| + SoFA | 20.5 | 15.0 | 25.6 | 0.0 | 15.3 |
| + Ours | 22.4 | 17.0 | 27.5 | 0.0 | 16.7 |
| Qwen-Base (Bai et al., 2023) | 10.0 | 8.0 | 15.0 | 0.0 | 8.3 |
| + VAR | 11.0 | 9.2 | 15.0 | 0.0 | 8.8 |
| + SoFA | 11.1 | 9.8 | 16.5 | 0.0 | 9.4 |
| + Ours | 13.6 | 11.4 | 19.7 | 0.0 | 11.2 |

Table 6: Experiment results on LLaVA-Interleave Bench (LIBench). SD: Spot the Difference, IE: Image Edit Instruction, VST: Visual Story Telling, TRVQA: Text-rich VQA, MIVQA: Multi-image VQA, QB: Q-Bench, SQ: ScanQA, Math: MathVerse-mv, Sci: SciVerse-mv.

| Model | SD | IE | VST | TRVQA | MIVQA | Puzzle |
|---|---|---|---|---|---|---|
| *Closed-source LMMs* | | | | | | |
| GPT-4o (OpenAI, 2023) | 14.2 | 12.3 | 12.0 | 58.7 | 55.6 | 18.9 |
| GPT-4V (Achiam et al., 2023) | 12.5 | 11.0 | 10.9 | 54.5 | 52.0 | 17.1 |
| Claude3.5 (Anthropic, 2023) | 13.1 | 11.4 | 11.5 | 56.1 | 53.3 | 17.9 |
| Gemini1.5 (Team et al., 2023) | 12.0 | 10.7 | 10.5 | 55.0 | 52.5 | 17.0 |

Table 6 – continued from previous page

| Model | SD | IE | VST | TRVQA | MIVQA | Puzzle |
|---|---|---|---|---|---|---|
| Gemini1.0 (Team et al., 2023) | 9.6 | 9.1 | 8.8 | 49.2 | 47.0 | 14.8 |
| *Multi-Image input LMMs* | | | | | | |
| Mantis-SigCLIP-8B (Jiang et al., 2024) | 17.6 | 11.2 | 12.5 | 45.2 | 52.5 | 25.7 |
| + VAR | 17.4 | 10.9 | 12.1 | 44.5 | 53.3 | 24.9 |
| + SoFA | 17.9 | 12.0 | 12.6 | 45.0 | 52.1 | 24.4 |
| + Ours | 21.1 | 14.4 | 15.2 | 49.7 | 57.3 | 28.1 |
| LLaVA-Interleave-7B (Li et al., 2025) | 37.1 | 24.3 | 33.1 | 76.1 | 87.5 | 48.7 |
| + VAR | 38.7 | 24.6 | 33.5 | 76.6 | 88.2 | 50.5 |
| + SoFA | 38.9 | 25.1 | 34.0 | 77.1 | 89.1 | 50.0 |
| + Ours | 40.9 | 27.8 | 36.4 | 79.5 | 90.7 | 52.8 |
| InternVL2-Pro (Chen et al., 2024) | 30.4 | 20.3 | 28.2 | 70.6 | 82.1 | 43.0 |
| + VAR | 30.8 | 20.5 | 28.3 | 70.0 | 81.4 | 44.1 |
| + SoFA | 31.4 | 21.0 | 28.9 | 71.6 | 83.0 | 44.5 |
| + Ours | 33.8 | 23.6 | 31.1 | 73.9 | 84.9 | 46.3 |
| InternVL1.5-chat (Chen et al., 2024) | 26.2 | 18.1 | 26.5 | 62.7 | 75.2 | 38.2 |
| + VAR | 27.0 | 17.8 | 27.0 | 63.1 | 75.5 | 38.3 |
| + SoFA | 27.8 | 18.2 | 27.2 | 63.8 | 76.0 | 39.7 |
| + Ours | 29.9 | 21.1 | 29.5 | 66.2 | 78.1 | 41.5 |
| InternVL2-8B (Chen et al., 2024) | 28.6 | 19.6 | 27.1 | 65.1 | 78.0 | 40.6 |
| + VAR | 28.8 | 20.0 | 29.2 | 64.6 | 75.1 | 40.8 |
| + SoFA | 29.6 | 20.3 | 28.0 | 66.2 | 79.4 | 42.0 |
| + Ours | 31.9 | 22.3 | 30.0 | 68.4 | 81.0 | 44.1 |
| Mini-InternVL-1.5-4B (Gao et al., 2024) | 22.4 | 14.8 | 22.7 | 56.2 | 69.3 | 33.6 |
| + VAR | 23.0 | 14.9 | 23.7 | 57.0 | 70.6 | 34.0 |
| + SoFA | 23.9 | 16.1 | 24.0 | 57.7 | 71.0 | 34.9 |
| + Ours | 25.8 | 17.9 | 25.8 | 59.7 | 72.8 | 36.9 |
| Mini-InternVL-1.5-2B (Gao et al., 2024) | 20.1 | 13.2 | 20.5 | 52.4 | 66.7 | 31.2 |
| + VAR | 20.8 | 13.1 | 19.6 | 53.7 | 66.9 | 32.1 |
| + SoFA | 21.5 | 14.3 | 21.9 | 54.0 | 68.1 | 32.4 |
| + Ours | 21.3 | 16.1 | 23.7 | 55.9 | 69.8 | 34.3 |
| Idefics-9B-Instruct (Bai et al., 2023) | 12.1 | 9.4 | 12.5 | 31.4 | 45.1 | 19.4 |
| + VAR | 11.0 | 9.5 | 13.5 | 32.7 | 45.2 | 18.6 |
| + SoFA | 13.3 | 10.2 | 16.9 | 32.3 | 46.6 | 20.9 |
| + Ours | 15.1 | 12.1 | 15.8 | 34.6 | 48.3 | 22.8 |
| idefics2-8B (Laurençon et al., 2024) | 20.0 | 13.5 | 20.6 | 40.2 | 58.1 | 28.1 |
| + VAR | 20.2 | 14.0 | 20.7 | 40.6 | 57.2 | 28.2 |
| + SoFA | 21.5 | 14.3 | 21.9 | 41.1 | 59.6 | 29.5 |
| + Ours | 22.6 | 17.3 | 23.9 | 45.5 | 62.5 | 33.4 |
| DeepSeek-VL-7B (Lu et al., 2024) | 24.3 | 16.0 | 23.9 | 58.0 | 71.1 | 35.1 |
| + VAR | 25.5 | 12.0 | 22.2 | 57.3 | 70.4 | 30.4 |
| + SoFA | 21.2 | 17.4 | 24.7 | 59.0 | 72.7 | 36.7 |
| + Ours | 27.7 | 19.2 | 27.9 | 62.2 | 74.6 | 39.8 |
| XComposer2-7B (Dong et al., 2024) | 24.1 | 16.7 | 24.3 | 55.5 | 70.1 | 34.2 |
| + VAR | 25.0 | 16.9 | 24.4 | 53.9 | 70.6 | 32.5 |
| + SoFA | 25.0 | 17.4 | 25.8 | 56.6 | 71.1 | 35.9 |
| + Ours | 27.3 | 19.8 | 29.6 | 57.8 | 73.2 | 39.0 |
| DeepSeek-VL-1.3B (Lu et al., 2024) | 16.7 | 11.5 | 17.6 | 46.2 | 58.7 | 25.4 |
| + VAR | 17.0 | 11.5 | 16.7 | 45.6 | 57.8 | 25.6 |
| + SoFA | 17.9 | 12.2 | 19.1 | 47.2 | 60.1 | 26.9 |
| + Ours | 19.8 | 17.0 | 22.9 | 49.4 | 62.9 | 28.8 |

Table 6 – continued from previous page

| Model | SD | IE | VST | TRVQA | MIVQA | Puzzle |
|---|---|---|---|---|---|---|
| OpenFlamingo-v2 (Awadalla et al., 2023) | 15.0 | 10.4 | 15.6 | 42.7 | 56.3 | 23.8 |
| + VAR | 14.0 | 12.5 | 14.7 | 42.0 | 57.4 | 25.0 |
| + SoFA | 16.3 | 11.8 | 16.1 | 43.6 | 57.8 | 25.3 |
| + Ours | 19.1 | 16.4 | 18.2 | 47.8 | 59.6 | 28.1 |
| XComposer2-1.8B (Dong et al., 2024) | 21.0 | 14.4 | 21.6 | 52.2 | 67.5 | 31.0 |
| + VAR | 22.1 | 10.6 | 22.8 | 52.7 | 64.6 | 31.4 |
| + SoFA | 24.4 | 15.2 | 23.1 | 53.3 | 68.9 | 32.7 |
| + Ours | 24.3 | 19.1 | 24.9 | 55.6 | 70.4 | 35.6 |
| Qwen2-VL (Team, 2025) | 18.9 | 13.7 | 17.8 | 38.9 | 51.5 | 23.1 |
| + VAR | 21.0 | 13.5 | 18.9 | 39.0 | 52.3 | 24.5 |
| + SoFA | 20.8 | 15.0 | 19.5 | 40.2 | 52.7 | 25.1 |
| + Ours | 23.5 | 17.3 | 21.7 | 42.4 | 55.0 | 27.3 |
| Qwen-VL (Bai et al., 2023) | 13.0 | 10.0 | 13.7 | 35.0 | 49.4 | 21.0 |
| + VAR | 16.0 | 10.1 | 11.7 | 32.3 | 18.6 | 21.0 |
| + SoFA | 14.3 | 10.8 | 15.0 | 36.0 | 50.9 | 22.3 |
| + Ours | 18.2 | 16.7 | 18.9 | 38.2 | 52.6 | 24.2 |
| Qwen-Base (Bai et al., 2023) | 8.6 | 7.5 | 9.0 | 24.8 | 38.6 | 15.2 |
| + VAR | 7.6 | 9.6 | 11.0 | 24.1 | 36.8 | 17.3 |
| + SoFA | 9.3 | 8.3 | 10.4 | 25.8 | 40.2 | 16.6 |
| + Ours | 11.4 | 10.2 | 13.1 | 27.9 | 42.9 | 18.4 |

Table 6 - Experiment results on LLaVA-Interleave (QB to MMMU).

| Model | QB | NLVR | Math | Sci | Mantis | BLINK | MMMU |
|---|---|---|---|---|---|---|---|
| *Closed-source LMMs* | | | | | | | |
| GPT-4o (OpenAI, 2023) | 79.4 | 90.3 | 61.5 | 68.2 | 64.1 | 52.7 | 49.3 |
| GPT-4V (Achiam et al., 2023) | 76.5 | 88.8 | 60.3 | 66.9 | 62.7 | 51.1 | 47.9 |
| Claude3.5 (Anthropic, 2023) | 74.1 | 88.0 | 58.9 | 65.4 | 61.0 | 50.0 | 46.8 |
| Gemini1.5 (Team et al., 2023) | 72.0 | 87.5 | 57.6 | 64.0 | 59.5 | 48.8 | 45.5 |
| Gemini1.0 (Team et al., 2023) | 65.4 | 83.2 | 49.8 | 57.3 | 54.1 | 44.0 | 40.3 |
| *Multi-Image input LMMs* | | | | | | | |
| Mantis-idefics2-8B (Jiang et al., 2024) | 69.9 | 87.4 | 27.2 | 29.3 | 59.5 | 46.4 | 34.1 |
| + VAR | 70.1 | 86.6 | 27.6 | 27.7 | 60.8 | 45.7 | 35.2 |
| + SoFA | 70.8 | 88.1 | 28.1 | 30.2 | 61.2 | 48.0 | 35.5 |
| + Ours | 73.4 | 90.3 | 33.7 | 34.4 | 63.0 | 49.9 | 37.3 |
| Mantis-SigCLIP-8B (Jiang et al., 2024) | 67.5 | 86.0 | 26.0 | 28.0 | 57.8 | 45.0 | 32.8 |
| + VAR | 66.6 | 88.0 | 26.3 | 28.4 | 57.1 | 46.2 | 33.9 |
| + SoFA | 69.0 | 86.6 | 27.0 | 29.0 | 58.4 | 46.6 | 34.2 |
| + Ours | 71.2 | 88.5 | 29.1 | 31.1 | 61.1 | 48.4 | 36.0 |
| LLaVA-Interleave-7B (Li et al., 2025) | 74.2 | 88.8 | 32.8 | 31.6 | 62.7 | 52.6 | 34.5 |
| + VAR | 76.5 | 89.0 | 33.1 | 32.9 | 62.8 | 53.8 | 35.5 |
| + SoFA | 76.0 | 89.4 | 33.6 | 32.5 | 64.2 | 54.1 | 36.0 |
| + Ours | 78.2 | 90.2 | 37.9 | 34.4 | 66.0 | 57.0 | 38.9 |
| InternVL2-Pro (Chen et al., 2024) | 71.0 | 88.2 | 31.1 | 33.4 | 61.5 | 50.9 | 36.5 |
| + VAR | 70.1 | 86.5 | 30.2 | 34.6 | 62.5 | 52.1 | 37.7 |
| + SoFA | 72.4 | 89.0 | 31.9 | 34.2 | 63.0 | 52.5 | 37.9 |
| + Ours | 74.3 | 89.8 | 34.0 | 36.0 | 66.8 | 55.3 | 39.7 |
| InternVL1.5-chat (Chen et al., 2024) | 68.1 | 86.9 | 29.5 | 31.2 | 59.6 | 48.7 | 33.7 |
| + VAR | 69.0 | 88.1 | 28.8 | 32.0 | 60.8 | 49.9 | 34.8 |
| + SoFA | 69.7 | 87.7 | 30.3 | 32.1 | 61.1 | 50.3 | 35.1 |

Table 6 – continued from previous page

| Model | QB | NLVR2 | Math | Sci | Mantis | BLINK | MMMU |
|---|---|---|---|---|---|---|---|
| + Ours | 71.6 | 89.6 | 32.6 | 34.0 | 62.8 | 52.1 | 36.9 |
| InternVL2-8B (Chen et al., 2024) | 69.2 | 87.6 | 30.1 | 32.2 | 60.7 | 49.5 | 35.3 |
| + VAR | 70.0 | 86.8 | 30.3 | 31.4 | 60.9 | 50.7 | 34.4 |
| + SoFA | 70.7 | 88.4 | 31.0 | 33.0 | 62.2 | 51.1 | 36.7 |
| + Ours | 72.6 | 90.2 | 33.1 | 34.8 | 63.9 | 52.9 | 38.4 |
| Mini-InternVL-chat-1.5-4B (Gao et al., 2024) | 63.0 | 84.2 | 24.7 | 27.0 | 55.9 | 44.2 | 31.0 |
| + VAR | 64.0 | 84.5 | 24.9 | 27.3 | 56.1 | 45.0 | 32.7 |
| + SoFA | 64.5 | 85.1 | 25.6 | 27.9 | 57.4 | 45.7 | 32.4 |
| + Ours | 66.6 | 86.9 | 28.7 | 27.9 | 59.2 | 47.4 | 34.1 |
| Mini-InternVL-chat-1.5-2B (Gao et al., 2024) | 60.5 | 82.7 | 23.1 | 25.8 | 53.8 | 42.6 | 29.6 |
| + VAR | 61.0 | 83.0 | 24.3 | 27.0 | 55.0 | 43.7 | 30.7 |
| + SoFA | 62.0 | 83.5 | 23.9 | 26.7 | 55.3 | 44.1 | 31.0 |
| + Ours | 64.1 | 85.2 | 26.0 | 28.6 | 57.0 | 45.9 | 32.7 |
| idefics2-8B (Laurençon et al., 2024) | 58.7 | 80.9 | 22.0 | 24.2 | 51.6 | 41.0 | 28.2 |
| + VAR | 59.8 | 81.0 | 21.2 | 24.4 | 52.8 | 42.7 | 29.3 |
| + SoFA | 60.8 | 81.6 | 22.8 | 25.0 | 53.7 | 42.9 | 29.6 |
| + Ours | 62.2 | 83.4 | 24.9 | 26.8 | 54.9 | 45.2 | 33.2 |
| Idefics-9B-Instruct (Bai et al., 2023) | 49.4 | 73.6 | 18.3 | 19.6 | 44.0 | 36.1 | 24.1 |
| + VAR | 50.6 | 74.9 | 19.4 | 20.8 | 45.1 | 37.4 | 25.1 |
| + SoFA | 50.9 | 74.5 | 19.0 | 20.5 | 45.4 | 37.8 | 25.4 |
| + Ours | 53.1 | 76.6 | 21.2 | 22.4 | 47.2 | 39.6 | 27.0 |
| DeepSeek-VL-7B (Lu et al., 2024) | 66.1 | 85.1 | 27.3 | 29.6 | 58.3 | 46.8 | 33.2 |
| + VAR | 66.3 | 85.3 | 27.6 | 27.9 | 59.0 | 47.0 | 34.3 |
| + SoFA | 67.7 | 85.9 | 28.2 | 30.5 | 60.0 | 48.4 | 34.6 |
| + Ours | 69.8 | 87.7 | 30.4 | 32.4 | 61.7 | 50.2 | 36.3 |
| XComposer2-7B (Dong et al., 2024) | 68.7 | 86.7 | 28.4 | 30.6 | 59.0 | 47.5 | 33.8 |
| + VAR | 69.9 | 88.0 | 29.6 | 31.8 | 60.2 | 48.8 | 35.0 |
| + SoFA | 70.3 | 87.5 | 29.2 | 31.4 | 60.6 | 49.2 | 35.3 |
| + Ours | 72.4 | 89.4 | 31.3 | 33.3 | 62.3 | 51.0 | 37.0 |
| DeepSeek-VL-1.3B (Lu et al., 2024) | 53.6 | 77.4 | 19.8 | 21.5 | 47.6 | 38.0 | 25.7 |
| + VAR | 54.0 | 76.7 | 21.0 | 22.7 | 48.8 | 39.2 | 26.8 |
| + SoFA | 55.1 | 78.3 | 20.6 | 22.3 | 49.1 | 39.6 | 27.1 |
| + Ours | 57.3 | 80.1 | 22.8 | 24.2 | 50.9 | 41.4 | 28.8 |
| OpenFlamingo-v2 (Awadalla et al., 2023) | 50.4 | 75.0 | 18.0 | 20.1 | 45.7 | 36.6 | 24.2 |
| + VAR | 50.6 | 76.1 | 19.2 | 21.4 | 46.9 | 37.7 | 25.3 |
| + SoFA | 51.9 | 75.7 | 18.9 | 21.0 | 47.3 | 38.1 | 25.6 |
| + Ours | 54.0 | 77.6 | 21.0 | 22.9 | 49.0 | 39.9 | 27.3 |
| XComposer2-1.8B (Dong et al., 2024) | 62.8 | 83.6 | 22.6 | 24.7 | 52.9 | 43.5 | 29.2 |
| + VAR | 63.0 | 84.1 | 23.1 | 24.0 | 54.1 | 44.8 | 30.3 |
| + SoFA | 64.3 | 84.5 | 23.4 | 25.6 | 54.5 | 45.2 | 30.7 |
| + Ours | 66.4 | 86.3 | 25.6 | 27.6 | 56.2 | 47.0 | 32.4 |
| Qwen2-VL (Team, 2025) | 49.2 | 73.4 | 18.9 | 20.5 | 44.7 | 35.5 | 24.8 |
| + VAR | 51.6 | 73.0 | 19.5 | 21.0 | 45.5 | 36.6 | 24.5 |
| + SoFA | 51.2 | 74.7 | 20.2 | 22.0 | 46.0 | 37.0 | 26.2 |
| + Ours | 53.5 | 76.8 | 22.5 | 24.1 | 48.3 | 39.2 | 28.4 |
| Qwen-VL (Bai et al., 2023) | 47.2 | 70.5 | 16.7 | 18.5 | 42.3 | 34.0 | 22.7 |
| + VAR | 48.0 | 70.9 | 13.9 | 18.8 | 43.6 | 34.2 | 23.0 |
| + SoFA | 48.7 | 71.4 | 17.5 | 19.4 | 43.9 | 35.6 | 23.2 |
| + Ours | 50.8 | 73.2 | 19.7 | 21.3 | 45.7 | 37.4 | 25.9 |
| Qwen-Base (Bai et al., 2023) | 40.2 | 65.0 | 12.6 | 14.0 | 37.1 | 30.5 | 19.1 |

Table 6 – continued from previous page

| Model | QB | NLVR2 | Math | Sci | Mantis | BLINK | MMMU |
|-------|------|------|------|------|------|------|------|
| + VAR | 41.4 | 64.3 | 11.8 | 15.3 | 37.4 | 30.7 | 20.2 |
| + SoFA | 41.0 | 65.9 | 13.4 | 14.9 | 38.7 | 30.1 | 20.5 |
| + Ours | 44.0 | 67.7 | 15.6 | 16.9 | 40.5 | 33.9 | 22.1 |

Table 7: Experiment results on MIBench (Multi-Image Instruction). GC: General Comparison, SD: Subtle Difference, VR: Visual Referring, TR: Temporal Reasoning, LR: Logical Reasoning, FVR: Fine-grained Visual Recognition, VTK: Vision-linked Textual Knowledge, TVK: Text-linked Visual Knowledge.

| Model | GC | SD | VR | TR | LR |
|-------|------|------|------|------|------|
| *Closed-source LMMs* | | | | | |
| GPT-4o (OpenAI, 2023) | 80.7 | 90.5 | 46.8 | 68.0 | 69.8 |
| GPT-4V (Achiam et al., 2023) | 72.8 | 79.2 | 45.8 | 61.8 | 66.3 |
| Claude3.5 (Anthropic, 2023) | 77.2 | 86.4 | 46.2 | 66.1 | 68.0 |
| Gemini1.5 (Team et al., 2023) | 74.0 | 82.0 | 44.1 | 63.0 | 66.8 |
| Gemini1.0 (Team et al., 2023) | 65.2 | 73.5 | 40.0 | 56.2 | 60.1 |
| *Multi-Image input LMMs* | | | | | |
| Mantis (Jiang et al., 2024) | 83.0 | 54.1 | 37.6 | 45.5 | 63.4 |
| + VAR | 83.5 | 55.0 | 36.9 | 46.0 | 63.9 |
| + SoFA | 84.0 | 55.3 | 36.6 | 46.6 | 64.6 |
| + Ours | 86.6 | 58.1 | 40.8 | 49.4 | 66.9 |
| LLaVA-Interleave-7B (Li et al., 2025) | 68.4 | 50.3 | 35.1 | 42.6 | 60.2 |
| + VAR | 68.8 | 50.6 | 36.4 | 43.0 | 60.5 |
| + SoFA | 69.4 | 51.1 | 36.9 | 43.6 | 61.2 |
| + Ours | 71.7 | 53.7 | 39.1 | 46.8 | 64.3 |
| InternVL2-Pro (Chen et al., 2024) | 79.5 | 88.0 | 44.5 | 66.0 | 69.0 |
| + VAR | 79.9 | 88.3 | 43.9 | 67.4 | 69.3 |
| + SoFA | 80.5 | 88.4 | 45.4 | 66.9 | 69.9 |
| + Ours | 82.6 | 90.9 | 48.7 | 69.2 | 72.0 |
| InternVL1.5-chat (Chen et al., 2024) | 60.8 | 70.5 | 33.0 | 52.0 | 58.2 |
| + VAR | 62.0 | 71.7 | 33.2 | 51.2 | 58.5 |
| + SoFA | 61.7 | 71.9 | 33.8 | 52.9 | 59.1 |
| + Ours | 64.1 | 73.8 | 35.9 | 55.0 | 61.3 |
| InternVL2-8B (Chen et al., 2024) | 74.2 | 82.6 | 41.2 | 60.3 | 65.5 |
| + VAR | 74.6 | 82.8 | 40.4 | 60.6 | 63.7 |
| + SoFA | 75.2 | 83.4 | 41.9 | 61.1 | 66.3 |
| + Ours | 77.2 | 85.6 | 43.9 | 63.2 | 68.4 |
| Mini-InternVL-1.5-4B (Gao et al., 2024) | 52.0 | 58.0 | 28.0 | 40.5 | 50.0 |
| + VAR | 52.2 | 58.2 | 27.1 | 39.8 | 51.2 |
| + SoFA | 53.5 | 59.5 | 29.4 | 42.0 | 51.0 |
| + Ours | 55.7 | 61.7 | 31.3 | 43.9 | 53.4 |
| Mini-InternVL-1.5-2B (Gao et al., 2024) | 49.2 | 55.1 | 26.2 | 38.0 | 47.5 |
| + VAR | 50.4 | 56.0 | 25.3 | 38.2 | 47.9 |
| + SoFA | 50.1 | 56.0 | 26.0 | 38.8 | 48.3 |
| + Ours | 52.4 | 58.3 | 29.0 | 41.0 | 50.4 |
| Idefics2-8B (Laurençon et al., 2024) | 83.1 | 49.7 | 32.6 | 44.8 | 56.4 |
| + VAR | 83.2 | 50.0 | 31.8 | 45.2 | 56.6 |
| + SoFA | 83.8 | 50.6 | 33.2 | 45.8 | 57.2 |
| + Ours | 84.0 | 53.3 | 36.6 | 47.9 | 59.3 |
| Idefics-9B-Instruct (Bai et al., 2023) | 40.3 | 28.4 | 18.7 | 26.1 | 35.0 |

Table 7 – continued from previous page

| Model | GC | SD | VR | TR | LR |
|---|---|---|---|---|---|
| + VAR | 41.0 | 27.8 | 18.0 | 26.5 | 32.2 |
| + SoFA | 41.2 | 29.3 | 19.0 | 27.1 | 35.9 |
| + Ours | 44.6 | 32.0 | 24.3 | 29.8 | 38.4 |
| DeepSeek-VL-7B (Lu et al., 2024) | 58.7 | 64.0 | 31.5 | 48.5 | 55.1 |
| + VAR | 57.9 | 65.3 | 32.7 | 49.0 | 54.3 |
| + SoFA | 60.2 | 65.9 | 33.0 | 49.5 | 56.0 |
| + Ours | 62.4 | 67.2 | 35.1 | 51.7 | 58.5 |
| DeepSeek-VL-1.3B (Lu et al., 2024) | 45.0 | 50.2 | 22.0 | 36.2 | 46.0 |
| + VAR | 46.0 | 50.6 | 22.3 | 34.4 | 45.2 |
| + SoFA | 46.0 | 51.2 | 22.8 | 37.0 | 46.0 |
| + Ours | 48.6 | 53.8 | 25.0 | 39.3 | 49.0 |
| XComposer2-7B (Dong et al., 2024) | 62.5 | 67.8 | 34.0 | 50.0 | 57.2 |
| + VAR | 63.0 | 69.1 | 33.2 | 51.4 | 58.4 |
| + SoFA | 63.4 | 68.7 | 34.0 | 51.0 | 58.0 |
| + Ours | 65.6 | 71.3 | 36.9 | 54.2 | 60.1 |
| XComposer2-1.8B (Dong et al., 2024) | 50.4 | 56.0 | 23.8 | 38.7 | 48.3 |
| + VAR | 50.5 | 56.2 | 20.0 | 38.9 | 48.7 |
| + SoFA | 51.8 | 57.5 | 24.6 | 39.1 | 49.2 |
| + Ours | 53.9 | 59.6 | 26.7 | 41.6 | 51.1 |
| OpenFlamingo-v2 (Awadalla et al., 2023) | 38.1 | 30.5 | 17.0 | 25.2 | 34.0 |
| + VAR | 39.4 | 31.7 | 18.4 | 25.6 | 34.3 |
| + SoFA | 39.9 | 31.2 | 18.9 | 26.1 | 34.0 |
| + Ours | 41.6 | 33.6 | 22.1 | 28.3 | 37.0 |
| Qwen2-VL (Team, 2025) | 55.6 | 31.2 | 21.0 | 29.5 | 43.8 |
| + VAR | 56.9 | 32.5 | 22.3 | 30.6 | 44.9 |
| + SoFA | 57.3 | 33.0 | 22.9 | 31.2 | 45.4 |
| + Ours | 59.8 | 35.4 | 25.1 | 33.7 | 47.7 |
| Qwen-VL (Bai et al., 2023) | 45.9 | 22.5 | 16.3 | 27.5 | 36.8 |
| + VAR | 45.2 | 22.8 | 16.6 | 24.8 | 35.1 |
| + SoFA | 46.8 | 23.4 | 17.1 | 28.4 | 37.7 |
| + Ours | 49.1 | 25.9 | 19.5 | 30.9 | 40.0 |
| Qwen-Base (Bai et al., 2023) | 20.0 | 15.0 | 8.0 | 12.0 | 18.0 |
| + VAR | 21.0 | 14.2 | 8.3 | 10.2 | 16.2 |
| + SoFA | 21.1 | 15.0 | 8.9 | 12.9 | 18.9 |
| + Ours | 24.0 | 18.3 | 11.5 | 15.4 | 26.3 |

Table 7 - Experiment results on MIBench (Multimodal Knowledge-Seeking)

| Model | FVR | TRI | VTK | TVK |
|---|---|---|---|---|
| *Closed-source LMMs* | | | | |
| GPT-4o (OpenAI, 2023) | 98.3 | 74.8 | 54.7 | 63.3 |
| GPT-4V (Achiam et al., 2023) | 90.2 | 71.0 | 52.0 | 56.0 |
| Claude3.5 (Anthropic, 2023) | 94.0 | 72.8 | 53.3 | 58.8 |
| Gemini1.5 (Team et al., 2023) | 92.1 | 70.1 | 51.0 | 57.2 |
| Gemini1.0 (Team et al., 2023) | 88.0 | 66.5 | 48.0 | 54.0 |
| *Multi-Image input LMMs* | | | | |
| Mantis (Jiang et al., 2024) | 16.4 | 37.7 | 26.4 | 41.7 |
| + VAR | 17.0 | 37.0 | 27.0 | 42.1 |
| + SoFA | 17.1 | 38.5 | 27.1 | 42.7 |
| + Ours | 19.4 | 41.1 | 29.3 | 45.2 |

Table 7 – continued from previous page

| Model | FVR | TRI | VTK | TVK |
|---|---|---|---|---|
| LLaVA-Interleave-7B (Li et al., 2025) | 70.2 | 55.6 | 37.5 | 45.8 |
| + VAR | 70.9 | 55.9 | 37.7 | 46.2 |
| + SoFA | 71.0 | 56.4 | 38.2 | 46.8 |
| + Ours | 73.2 | 58.9 | 42.5 | 48.9 |
| InternVL2-Pro (Chen et al., 2024) | 85.6 | 63.5 | 47.0 | 52.8 |
| + VAR | 86.7 | 62.7 | 44.2 | 52.1 |
| + SoFA | 86.0 | 64.3 | 47.7 | 53.7 |
| + Ours | 88.1 | 66.6 | 49.9 | 55.9 |
| InternVL1.5-chat (Chen et al., 2024) | 78.0 | 59.2 | 42.1 | 49.0 |
| + VAR | 79.0 | 60.5 | 41.2 | 47.3 |
| + SoFA | 78.8 | 60.0 | 42.8 | 49.9 |
| + Ours | 81.0 | 62.4 | 44.9 | 52.1 |
| InternVL2-8B (Chen et al., 2024) | 82.4 | 61.0 | 45.3 | 51.0 |
| + VAR | 83.0 | 62.3 | 45.5 | 52.3 |
| + SoFA | 83.1 | 60.8 | 46.0 | 51.9 |
| + Ours | 85.0 | 64.0 | 48.1 | 53.9 |
| Mini-InternVL-1.5-4B (Gao et al., 2024) | 60.3 | 48.0 | 30.5 | 40.2 |
| + VAR | 61.0 | 49.3 | 31.7 | 41.5 |
| + SoFA | 61.1 | 48.8 | 31.2 | 41.1 |
| + Ours | 63.4 | 51.2 | 33.3 | 43.1 |
| Mini-InternVL-1.5-2B (Gao et al., 2024) | 57.8 | 45.6 | 28.4 | 37.9 |
| + VAR | 59.0 | 45.8 | 27.6 | 38.1 |
| + SoFA | 58.0 | 46.4 | 29.1 | 38.0 |
| + Ours | 60.8 | 48.8 | 33.2 | 40.8 |
| Idefics-9B-Instruct (Bai et al., 2023) | 48.6 | 40.1 | 23.6 | 32.0 |
| + VAR | 49.0 | 40.3 | 23.8 | 31.4 |
| + SoFA | 49.5 | 40.9 | 24.3 | 33.0 |
| + Ours | 52.2 | 43.5 | 26.6 | 35.4 |
| Idefics2-8B (Laurençon et al., 2024) | 42.4 | 43.9 | 25.6 | 39.0 |
| + VAR | 43.7 | 44.2 | 23.8 | 40.2 |
| + SoFA | 44.3 | 44.8 | 26.3 | 39.8 |
| + Ours | 48.8 | 47.1 | 28.5 | 41.9 |
| DeepSeek-VL-1.3B (Lu et al., 2024) | 55.2 | 42.0 | 26.0 | 35.1 |
| + VAR | 56.4 | 43.3 | 27.1 | 36.4 |
| + SoFA | 56.0 | 42.8 | 26.7 | 36.0 |
| + Ours | 58.1 | 45.0 | 28.8 | 38.0 |
| DeepSeek-VL-7B (Lu et al., 2024) | 62.0 | 50.1 | 31.9 | 41.6 |
| + VAR | 61.2 | 51.4 | 32.1 | 40.9 |
| + SoFA | 62.8 | 51.0 | 32.6 | 42.5 |
| + Ours | 65.0 | 53.4 | 34.7 | 44.7 |
| XComposer2-7B (Dong et al., 2024) | 64.6 | 52.3 | 33.7 | 43.0 |
| + VAR | 65.0 | 51.6 | 33.8 | 44.3 |
| + SoFA | 65.4 | 53.1 | 34.4 | 43.9 |
| + Ours | 67.6 | 55.4 | 36.5 | 45.9 |
| OpenFlamingo-v2 (Awadalla et al., 2023) | 46.0 | 34.8 | 19.8 | 28.7 |
| + VAR | 47.2 | 34.0 | 20.0 | 27.9 |
| + SoFA | 46.0 | 35.6 | 20.6 | 29.5 |
| + Ours | 49.1 | 37.9 | 22.9 | 31.6 |
| XComposer2-1.8B (Dong et al., 2024) | 58.7 | 46.1 | 29.6 | 39.8 |
| + VAR | 58.8 | 44.4 | 30.0 | 40.1 |

Table 7 – continued from previous page

| Model | FVR | TRI | VTK | TVK |
|---|---|---|---|---|
| + SoFA | 59.5 | 47.0 | 30.3 | 40.7 |
| + Ours | 61.6 | 49.3 | 32.4 | 42.6 |
| Qwen2-VL (Team, 2025) | 63.2 | 41.0 | 27.6 | 33.5 |
| + VAR | 64.5 | 42.2 | 28.7 | 34.1 |
| + SoFA | 65.0 | 42.7 | 29.1 | 34.6 |
| + Ours | 67.2 | 45.0 | 31.5 | 36.9 |
| Qwen-VL (Bai et al., 2023) | 58.8 | 35.9 | 22.9 | 18.1 |
| + VAR | 60.0 | 35.1 | 26.1 | 18.3 |
| + SoFA | 59.7 | 36.7 | 23.7 | 18.9 |
| + Ours | 62.0 | 39.2 | 25.9 | 21.1 |
| Qwen-Base (Bai et al., 2023) | 40.0 | 28.0 | 15.0 | 20.0 |
| + VAR | 40.6 | 28.3 | 16.0 | 19.3 |
| + SoFA | 41.1 | 28.9 | 15.9 | 20.9 |
| + Ours | 44.2 | 31.6 | 18.1 | 23.2 |

