# OpenReview forum: "Multiple Images Distract Large Multimodal Models via Attention Fragmentation"
_ICLR.cc/2026/Conference — ICLR 2026 Conference Withdrawn Submission_

### Official Review · Reviewer_AqZJ · 2025-10-31

**Soundness:** 3
**Presentation:** 3
**Contribution:** 3
**Rating:** 6
**Confidence:** 4

**Summary:**

This paper investigates the limitations of current Large Multimodal Models (LMMs) in understanding multiple images and introduces a novel concept called Attention Fragmentation.
Through empirical analysis, the authors reveal that when multiple images are processed in a single autoregressive sequence, attention tends to concentrate on similar background regions (so-called attention sinks) across images, especially for earlier ones due to causal masking. This leads to dispersed, low-utility attention and weak cross-image integration.

To address this, the authors propose Attention Remasking (AR)  to the pre-softmax attention scores. AR masks out sink tokens column-wise and selectively un-masks task-relevant cross-image tokens based on patch-level CLIP relevance guided by GroundingDINO. The freed attention is redistributed to these semantically meaningful links, preserving text autoregression.

Experiments demonstrate consistent improvements across various open-source LMMs. Analyses further show that AR reduces attention fragmentation, mitigates recency bias, and enhances cross-image reasoning without retraining.

**Strengths:**

- The paper identifies attention fragmentation as a fundamental limitation in multi-image LMMs — connecting visual attention sinks with causal masking and positional bias. This provides a clear and interpretable explanation for why models struggle with multi-image reasoning.

- The proposed Attention Remasking is a lightweight, training-free approach that can be universally applied to existing LMMs. It effectively reuses the model’s existing structure without introducing new parameters or requiring fine-tuning.

- Evaluations on diverse benchmarks and models show consistent performance gains.

- The paper is well-structured and easy to follow, with clear figures and thorough theoretical reasoning.

**Weaknesses:**

- AR relies on GroundingDINO and CLIP for patch-level relevance estimation, which introduces external biases and limits reproducibility. The performance may degrade if the grounding detector is weak or misaligned with the model’s visual encoder.

- Editing pre-softmax attention at inference could increase computational cost, but the paper does not report runtime or memory overhead.

- The paper focuses training-free methods. Comparisons with fine-tuned or retrained multi-image models would better contextualize its practical competitiveness.

**Questions:**

- How sensitive is AR to the accuracy or granularity of the grounding model (e.g., using weaker detectors or fewer regions)?

- Could the CLIP-based relevance prior be replaced by an internal attention signal from the model itself (to reduce reliance on external models)?

- Would fine-tuning a model with AR-modified attention patterns produce further gains, or does AR mainly serve as a post-hoc patch?

---

### Official Review · Reviewer_42zK · 2025-10-31

**Soundness:** 2
**Presentation:** 3
**Contribution:** 3
**Rating:** 2
**Confidence:** 5

**Summary:**

The paper argues that large multimodal models that accept several images at once get stuck attending to the same background patches in every image. Because attention is causal, earlier images attract even more wasteful attention, so the model fails to connect evidence across images. They call this attention fragmentation. The fix is attention remasking. They edit the pre softmax attention scores only in the visual to visual block, hard mask columns that correspond to sink tokens, then selectively open forward links from earlier queries to later image tokens when a text grounded patch relevance prior says those links matter. The freed attention budget is reassigned to those links. They claim this lowers attention dispersion across images and reduces order bias, improving accuracy on recent multi image benchmarks without any retraining.

**Strengths:**

1. The paper ties order sensitivity and flaky multi-image reasoning to a concrete mechanism (repeated sinks + causal masking), not just vibes. The Chamfer/entropy analyses and sink-share skew give a coherent story.
2. AR edits scores, not weights; preserves text autoregression; and uses grounded, instruction-conditioned priors to decide which forward links to unmask. That’s practical and model-agnostic.
3.They explain why VAR/SoFA fail conceptually (no new forward links; proportional redistribution preserves dispersion; untrained logits everywhere) and then show AR actually lowers late-layer entropy and reduces order bias.
4.Mask-only and unmask-only both underperform full AR; grounded-CLIP scoring outperforms uniform / naïve variants. That’s the kind of sanity check many papers skip.

**Weaknesses:**

1. The whole approach leans on a separate detector and CLIP to decide what is relevant. When grounding is brittle or the instruction is abstract, the method inherits those mistakes. There is no robustness story beyond trusting the detector. This is an external crutch, not an internal fix.
2. The routing policy is hand crafted. You reclaim a fixed sink budget and pour it into a sparse set of keys according to a similarity prior. There is no guarantee those queries want that mass and no stability or sensitivity analysis to show the edit cannot go sideways. It is a clever hack, but still a hack.
3.Selective unmasking mixes trained and never trained pathways. The paper itself points out that SoFA opens links that were never trained and performs unevenly. Here you also open untrained links, just fewer of them. There is no stress test for cases where CLIP overconfidently highlights the wrong region or where instructions do not map cleanly to patches.
4. The evaluation story avoids cost and failure analyses. We see accuracy bumps and nicer entropy curves, but no latency or memory breakdown from running a detector plus CLIP plus mask rewrites, no adversarial counter cases, and no tasks where pushing attention forward is the wrong move. The claims are broad and the tests are narrow

**Questions:**

1. What happens when grounding is wrong or the instruction is abstract or aesthetic. Does the method confidently shove attention into bad regions and lock in errors. Please quantify failure rates under degraded grounding.
2. How far can a patch level similarity prior carry genuine multi image reasoning like counts, geometry, and relational chains instead of just saliency chasing. Where does it break.
3. Does the edit ever interfere with long mixed prompts where text needs steady internal routing. Show text only and text heavy mixed tasks to prove there is no collateral damage.
4. Could a later image with high CLIP like regions hijack the freed budget and force order flips in the opposite direction. Any robustness checks against adversarial sinks and spurious relevance.
5. What are the true compute and latency costs at realistic batch sizes and image counts. Do the gains hold under tight inference budgets or are we trading accuracy for throughput.

---

### Official Review · Reviewer_1bgt · 2025-11-01

**Soundness:** 2
**Presentation:** 2
**Contribution:** 3
**Rating:** 4
**Confidence:** 4

**Summary:**

The paper investigates a crucial issue in Large Multimodal Models (LMMs) that handle multiple images, where the models suffer from attention fragmentation. To solve this, the authors propose a method called Attention Remasking (AR), which is a post-training technique that modifies the attention score matrix. This method masks "sink" tokens and unmasks task-relevant cross-image tokens, improving multi-image understanding without requiring retraining or hyperparameter tuning. The results show that AR mitigates attention fragmentation and improves performance on multiple image benchmarks.

**Strengths:**

1. The identification of attention fragmentation in multi-image LMMs and the introduction of AR to address this issue is a novel contribution.
2. The experimental methodology is sound, using well-defined benchmarks and appropriate metrics. The empirical evidence supports the claims of the paper, showing that AR reduces attention fragmentation and improves model performance.
3. The method presented has the potential to impact the way multimodal models are designed, improving their accuracy and efficiency in tasks that require multi-image understanding.

**Weaknesses:**

1. While the paper performs extensive testing on well-established multi-image benchmarks like MMIU, MuirBench, and MIRB, the results remain largely confined to these curated datasets. These benchmarks are useful, but they may not represent the full range of challenges present in real-world multimodal tasks. Real-world images often come with more noise, more complex interactions between images, and varied content that may require a different approach to attention allocation. For instance, highly dynamic images (e.g., images involving fast-moving objects or complex scenes) or images with low resolution might not behave the same way as those in the current benchmarks. Additionally, considering adversarially trained datasets such as the GTSRB dataset [1] is also an option.
2. One of the strengths of AR is its post-training application, but the computational complexity of the technique remains unclear. The paper does not discuss the scalability of AR, especially in the context of large models or very large datasets. As LMMs continue to grow in size (think of models with billions of parameters), the overhead introduced by manipulating attention matrices post-training could become significant.
3. While the paper shows that AR improves multi-image understanding, it doesn’t provide a comprehensive ablation study. The authors mention DINO-grounded CLIP score/Masking & Unmasking visual tokens, but they do not break down the impact of each individual component on model performance. Is the core improvement due to DINO-grounded CLIP score, or masking attention sinks & unmasking cross-image links? Given that AR is a post-training edit, understanding which component has the most significant effect would give future researchers a clearer path for improvement.
4. The paper’s method, AR, focuses on the manipulation of attention weights based on a grounded patch relevance score. While this appears to be effective for the benchmarks, it raises concerns about overfitting the model to certain types of images. Specifically, the reliance on CLIP-based grounding may make the model too dependent on image content that aligns with the predefined semantic concepts of CLIP. When the images do not fit well into these predefined concepts, or when the grounded relevance does not accurately represent task-relevant information, AR's performance will drop.

[1] Stallkamp, J., Schlipsing, M., Salmen, J., & Igel, C. (2011, July). The German traffic sign recognition benchmark: a multi-class classification competition. In The 2011 international joint conference on neural networks (pp. 1453-1460). IEEE.

**Questions:**

1. The current benchmarks used in the paper do not reflect the diversity of real-world images, especially those with poor quality or significant noise. In practical applications like surveillance or social media analysis, the model will often have to process images that may be blurry, low-resolution, or distorted. Would AR still be effective in such cases, or does the technique rely too heavily on high-quality, well-structured images for successful attention manipulation?
2. One of the key benefits of the AR method is that it does not require retraining the model. However, what impact does this post-processing step have on real-time performance? In applications where speed is crucial, such as autonomous driving or real-time video analysis, could the time spent remasking attention and recalculating attention scores become a bottleneck?
3. The current results show that AR improves multi-image accuracy and reduces recency bias. However, over the long term, does it cause any catastrophic forgetting of previously learned behaviors when the model encounters unseen data distributions [2]?

[2] Yang, J., Wang, P., Zou, D., Zhou, Z., Ding, K., Peng, W., ... & Liu, Z. (2022). Openood: Benchmarking generalized out-of-distribution detection. Advances in Neural Information Processing Systems, 35, 32598-32611.

---

### Official Review · Reviewer_FiPB · 2025-11-01

**Soundness:** 3
**Presentation:** 3
**Contribution:** 2
**Rating:** 6
**Confidence:** 3

**Summary:**

The paper studies multi-image LMMs, diagnosing attention “fragmentation” and recency bias, and proposes a post-training Attention Remasking (AR) that masks visual sink tokens and selectively un-masks cross-image links to re-allocate attention. It reports consistent gains across several open-source LMMs with simple plug-in changes.

**Strengths:**

1. Clearly formulates an important failure mode (fragmentation/recency) with intuitive metrics and diagnostics.
2. Method is simple, training-free, and targeted (operates on visual–visual attention only), with informative ablations.
3. Broad empirical coverage across multiple LMMs/datasets showing consistent improvements.

**Weaknesses:**

1. “Large Multmodal Model (LMM)” (missing “i”) in Preliminaries.
2. Eq. (8) is “defined when w(i,ℓ)>0”, but the manuscript does not state how entropy and subsequent statistics are handled when 𝑤(𝑖,ℓ)=0 (e.g., skip, impute, or add 𝜖). This affects fragmentation estimates and hypothesis tests.
3. The method “relaxes the visual parts of "𝑀_causal” based on relevance, but there is no formal constraint to prevent unintended information flow (e.g., unmasking transitive chains across later images) nor analysis of complexity/sparsity guarantees for the edited mask. Provide a clear rule or bound for which visual links can be unmasked and why leakage cannot occur.
4. You report detector thresholds and a 0.1–0.9 threshold sweep; what’s missing is end-to-end runtime/throughput and memory overhead of AR per model/dataset, a proposal-count ablation (e.g., top-K), complexity stats for the edited mask (avg nonzeros per row), and a brief detector/backbone swap study. Include cost-vs-accuracy plots to support practical adoption.
5. Beyond Wilcoxon+Holm and the Hodges–Lehmann median shift, please add per-task effect sizes with 95% CIs, seed variance (mean±std over runs), and per-dataset tables of absolute accuracy deltas. This will clarify practical significance and guard against over-generalizing pooled results.
6. Experiments focus on accuracy/entropy and order sensitivity; missing are robustness tests under noisy/over-segmented proposals, occlusion, or multi-turn prompts where mask edits persist across turns. Add these to demonstrate stability beyond single-turn benchmarks.

**Questions:**

1. “Large Multmodal Model (LMM)” (missing “i”) in Preliminaries.
2. Eq. (8) is “defined when w(i,ℓ)>0”, but the manuscript does not state how entropy and subsequent statistics are handled when 𝑤(𝑖,ℓ)=0 (e.g., skip, impute, or add 𝜖). This affects fragmentation estimates and hypothesis tests.
3. The method “relaxes the visual parts of "𝑀_causal” based on relevance, but there is no formal constraint to prevent unintended information flow (e.g., unmasking transitive chains across later images) nor analysis of complexity/sparsity guarantees for the edited mask. Provide a clear rule or bound for which visual links can be unmasked and why leakage cannot occur.
4. You report detector thresholds and a 0.1–0.9 threshold sweep; what’s missing is end-to-end runtime/throughput and memory overhead of AR per model/dataset, a proposal-count ablation (e.g., top-K), complexity stats for the edited mask (avg nonzeros per row), and a brief detector/backbone swap study. Include cost-vs-accuracy plots to support practical adoption.
5. Beyond Wilcoxon+Holm and the Hodges–Lehmann median shift, please add per-task effect sizes with 95% CIs, seed variance (mean±std over runs), and per-dataset tables of absolute accuracy deltas. This will clarify practical significance and guard against over-generalizing pooled results.
6. Experiments focus on accuracy/entropy and order sensitivity; missing are robustness tests under noisy/over-segmented proposals, occlusion, or multi-turn prompts where mask edits persist across turns. Add these to demonstrate stability beyond single-turn benchmarks.

---

### Note · Authors · 2025-12-04

I have read and agree with the venue's withdrawal policy on behalf of myself and my co-authors.